# Genome-wide association study of depression phenotypes in UK Biobank identifies variants in excitatory synaptic pathways

David M. Howard [1], Mark J. Adams [1], Masoud Shirali [1], Toni-Kim Clarke[1], Riccardo E. Marioni[2], Gail Davies[2,3], Jonathan R.I. Coleman [4,5], Clara Alloza[1], Xueyi Shen [1], Miruna C. Barbu[1], Eleanor M. Wigmore[1], Jude Gibson[1], 23andMe Research Team, Saskia P. Hagenaars[4,5], Cathryn M. Lewis [4,5], Joey Ward[6], Daniel J. Smith [6], Patrick F. Sullivan[7,8,9], Chris S. Haley [10], Gerome Breen[4,5], Ian J. Deary[2,3] & Andrew M. McIntosh [1,3]

Depression is a polygenic trait that causes extensive periods of disability. Previous genetic studies have identified common risk variants which have progressively increased in number with increasing sample sizes of the respective studies. Here, we conduct a genome-wide association study in 322,580 UK Biobank participants for three depression-related phenotypes: broad depression, probable major depressive disorder (MDD), and International Classification of Diseases (ICD, version 9 or 10)-coded MDD. We identify 17 independent loci that are significantly associated ($P < 5 \times 10^{-8}$) across the three phenotypes. The direction of effect of these loci is consistently replicated in an independent sample, with 14 loci likely representing novel findings. Gene sets are enriched in excitatory neurotransmission, mechanosensory behaviour, post synapse, neuron spine and dendrite functions. Our findings suggest that broad depression is the most tractable UK Biobank phenotype for discovering genes and gene sets that further our understanding of the biological pathways underlying depression.

[1] Division of Psychiatry, University of Edinburgh, Edinburgh EH10 5HF, UK. [2] Centre for Cognitive Ageing and Cognitive Epidemiology, University of Edinburgh, Edinburgh EH8 9JZ, UK. [3] Department of Psychology, University of Edinburgh, Edinburgh EH8 9JZ, UK. [4] Social Genetic and Developmental Psychiatry Centre, Institute of Psychiatry, Psychology & Neuroscience, King's College London, London SE5 8AF, UK. [5] NIHR Biomedical Research Centre for Mental Health, South London and Maudsley NHS Trust, London SE5 8AF, UK. [6] Institute of Health and Wellbeing, University of Glasgow, Glasgow G12 8RZ, UK. [7] Department of Medical Epidemiology and Biostatistics, Karolinska Institutet, Stockholm 171 77, Sweden. [8] Department of Genetics, University of North Carolina, Chapel Hill, 27599 NC, USA. [9] Department of Psychiatry, University of North Carolina, Chapel Hill, 27599 NC, USA. [10] Medical Research Council Human Genetics Unit, Institute of Genetics and Molecular Medicine, University of Edinburgh, Edinburgh EH4 2XU, UK  Correspondence and requests for materials should be addressed to D.M.H. (email: D.Howard@ed.ac.uk)
#A full list of consortium members appears at the end of the paper.

D epression is ranked as the largest contributor to global disability affecting 322 million people[1]. The heritability ($h^2$) of major depressive disorder (MDD) is estimated at 37% from twin studies[2], and common single nucleotide polymorphisms (SNPs) contribute ~9% to variation in liability[3], providing strong evidence of a genetic contribution to its causation. Previous genetic association studies have used a number of depression phenotypes, including self-declared depression[4], depressive symptoms[5], clinician diagnosed MDD[6] and depression ascertained via hospital records[7]. The favouring of greater sample size over clinical precision has yielded a steady increase over time in the number of variants for ever more diverse depression-related phenotypes[3,4,6,8]. Differing definitions of depression are rarely available within large studies, however, UK Biobank is a notable exception.

The UK Biobank cohort has been extensively phenotyped, allowing us to derive three depression traits: self-reported past help-seeking for problems with "nerves, anxiety, tension or depression" (termed "broad depression"), self-reported depressive symptoms with associated impairment (termed "probable MDD") and MDD identified from International Classification of Diseases (ICD)-9 or ICD-10-coded hospital admission records (termed "ICD-coded MDD"). The broad depression phenotype is likely to incorporate a number of personality and psychiatric disorders, whereas the probable MDD and ICD-coded MDD potentially offer more robust definitions for depression.

The UK Biobank cohort provides data on over 500,000 individuals and represents an opportunity to conduct a large genome-wide association study (GWAS) of depression, followed by replication of the significant loci in an independent cohort. We calculated the $h^2$ of the three UK Biobank definitions of depression and the genetic correlations between them. We also calculated the genetic correlations between the three depression phenotypes and other psychiatric disorders and a range of disease traits. We conducted further analyses to identify genes, regions, gene sets and tissues associated with each phenotype and used GTEx[9] to identify significant variants that were expression quantitative trait loci (eQTL). This approach allowed us to identify replicable loci associated with depression which were enriched across a number of plausible genes and gene sets involved in synaptic pathways.

## Results

**Genome-wide association study of depression.** We conducted a genome-wide association study testing the effect of 7,666,894 variants on three depression phenotypes (broad depression, probable MDD and ICD-coded MDD) using up to 322,580 UK Biobank participants. Broad depression was based on self-reported help-seeking behaviour for mental health difficulties from either a general practitioner or psychiatrist. Probable MDD was based on the Smith et al.[10] definition of MDD studied previously in UK Biobank. ICD-coded MDD was based on either a primary or secondary diagnosis of MDD in hospital admission records. The study demographics for the case and control groups within each UK Biobank phenotype are provided in Supplementary Table 1. A total of 17 independent variants were genome-wide significant ($P < 5 \times 10^{-8}$) across the three depression phenotypes analysed (Table 1), of which 14 were associated with broad depression, two were associated with probable MDD and one was associated with ICD-coded MDD. The effect sizes, standard errors and $P$-values for the associated variants across the three phenotypes are provided in Supplementary Data 1, with odds ratios calculated using a logistic regression model in Plink v1.90b4[11]. Replication of the 17 associated variants was sought using the association study of depression performed using

research participants from the personal genetics company 23andMe, Inc., conducted by Hyde et al[4]. There were 16 variants that had an effect in the same direction as the 23andMe analysis, with seven variants shown to be significant ($P < 0.0029$) in the 23andMe cohort after applying a Bonferroni correction (Table 1). All 17 variants remained significant ($P < 5 \times 10^{-8}$) in the meta-analysis. Fourteen out of the 17 significant variants identified within our analysis of UK Biobank were novel, i.e. not reported within ±500 Kb of a significant variant ($P < 5 \times 10^{-8}$) reported by either Hyde et al.[4], Okbay et al.[5], or the Converge Consortium[7]. The three variants (rs6699744, rs6424532 and rs1021363) that were unlikely to be novel were close to variants identified by Hyde et al[4].

Manhattan plots of all the variants analysed in UK Biobank are provided in Figs. 1, 2 and 3 for broad depression, probable MDD and ICD-coded MDD, respectively. Q–Q plots of the observed $P$-values on those expected are provided in Supplementary Figs. 1, 2, and 3 for broad depression, probable MDD and ICD-coded MDD, respectively. There were 3,690 variants with $P < 10^{-6}$ for an association with broad depression (Supplementary Data 2), 189 for probable MDD (Supplementary Data 3), and 107 for ICD-coded MDD (Supplementary Data 4). None of the phenotypes examined provided evidence of inflation of the test statistics due to population stratification, with any inflation due to polygenic signal (intercepts, standard errors and genomic inflation factors for each phenotype, calculated using Linkage Disequilibrium Score Regression (LDSR), are provided in Supplementary Table 2). Regional visualisation plots of rs3807865, rs1021363 and rs10501696 are provided in Supplementary Figs. 4, 5, and 6, respectively.

**Heritability of depression.** The SNP-based $h^2$ estimates on the liability scale for the whole sample using LDSR were 10.2% (±0.4%), 4.7% (±0.6%) and 10.0% (±1.2%) for broad depression, probable MDD and ICD-coded MDD, respectively. We estimated the SNP-based $h^2$ on the liability scale for each depression phenotype in UK Biobank using GCTA-GREML[12] within each recruitment centre and also by geographical region (Supplementary Tables 3, 4 and Fig. 4 (broad depression)). The SNP-based $h^2$ estimates for each region ranged from 8.4% to 17.8% for broad depression, 0% to 27.5% for probable MDD and from 0% to 25.4% for ICD-coded MDD. There was no evidence of heterogeneity between geographical regions for broad depression (I-squared = 0%, $P = 0.58$), probable MDD (I-squared = 33.1%, $P = 0.18$) or ICD-coded MDD (I-squared = 0%, $P = 0.69$).

**Genetic correlations with depression.** Strong genetic correlations ($r_g$) were found between each of the three UK Biobank depression phenotypes (0.85 (±0.05) ≤ $r_g$ ≤ 0.87 (±0.05), $P \leq 4.21 \times 10^{-59}$) using LDSR[13] (Supplementary Table 5). There were also significant differences ($P < 0.003$) between each of the three phenotypes when assessing whether $r_g = 1$.

LD Hub[14] was then used to calculate the genetic correlations between each depression phenotype and 235 other traits (Supplementary Data 5). After applying a false discovery rate correction there were 42, 31 and 35 significant correlations with other traits ($P_{FDR} < 0.05$) for broad depression, probable MDD and ICD-coded MDD, respectively. Significantly genetic correlations were observed between the UK Biobank depression-related phenotypes and the results of an analysis of clinically defined MDD[6] (broad depression ($r_g = 0.79 \pm 0.07$, $P_{FDR} = 3.79 \times 10^{-25}$), probable MDD ($r_g = 0.64 \pm 0.12$, $P_{FDR} = 7.95 \times 10^{-6}$) and ICD-coded MDD ($r_g = 0.63 \pm 0.10$, $P_{FDR} = 1.86 \times 10^{-8}$)). LD Hub[14] also identified significant genetic correlations between the three

**Table 1 Independent variants with a genome-wide significant ($P < 5 \times 10^{-8}$) association with broad depression, probable major depressive disorder (MDD) or International Classification of Diseases (ICD)-coded MDD in UK Biobank**

| Phenotype | Chr | Marker name | Position | A1/ A2 | UK Biobank | | | | | 23andMe | | Meta-analysis | | Direction |
|---|---|---|---|---|---|---|---|---|---|---|---|---|---|---|
| | | | | | Freq | Info | Beta (St. err) | Gene +/− 10 kb | P-value | Beta (St. err.) | P-value | Beta (St. err.) | P-value | |
| Broad depression | 1 | rs10127497 | 67050144 | T/A | 0.138 | 1.00 | 0.0097 (0.0017) | SGIP1 | $1.29 \times 10^{-8}$ | 0.0098 (0.0086) | 0.257 | 0.0097 (0.0017) | $6.63 \times 10^{-9}$ | ++ |
| | 1 | rs6699744 | 72825144 | T/A | 0.612 | 1.00 | 0.0089 (0.0012) | — | $1.64 \times 10^{-13}$ | 0.0328 (0.0064) | $2.68 \times 10^{-7}$ | 0.0098 (0.0012) | $2.29 \times 10^{-16}$ | ++ |
| | 1 | rs6424532 | 73664022 | A/G | 0.486 | 1.00 | 0.0065 (0.0012) | — | $3.91 \times 10^{-8}$ | 0.0233 (0.006) | $1.14 \times 10^{-4}$ | 0.0071 (0.0012) | $8.71 \times 10^{-10}$ | ++ |
| | 1 | rs7548151 | 177026983 | A/G | 0.084 | 1.00 | 0.0125 (0.0021) | ASTN1 | $3.87 \times 10^{-9}$ | 0.006 (0.0104) | 0.560 | 0.0122 (0.0021) | $3.93 \times 10^{-9}$ | ++ |
| | 5 | rs40465 | 103981726 | G/T | 0.332 | 1.00 | 0.0078 (0.0013) | RP11-6N13.1 | $4.45 \times 10^{-10}$ | 0.0193 (0.0064) | $2.63 \times 10^{-3}$ | 0.0082 (0.0012) | $2.10 \times 10^{-11}$ | ++ |
| | 6 | rs3132685 | 29945949 | A/G | 0.130 | 1.00 | −0.0131 (0.0018) | — | $2.47 \times 10^{-13}$ | −0.0249 (0.0099) | 0.011 | −0.0134 (0.0018) | $1.97 \times 10^{-14}$ | −− |
| | 6 | rs112348907 | 73587953 | G/A | 0.296 | 1.00 | 0.0073 (0.0013) | — | $1.52 \times 10^{-8}$ | −0.0004 (0.0067) | 0.950 | 0.0071 (0.0013) | $2.96 \times 10^{-8}$ | +− |
| | 7 | rs3807865 | 12250402 | A/G | 0.412 | 1.00 | 0.0082 (0.0012) | TMEM106B | $7.28 \times 10^{-12}$ | 0.019 (0.0061) | $2.00 \times 10^{-3}$ | 0.0086 (0.0012) | $2.55 \times 10^{-13}$ | ++ |
| | 7 | rs2402273 | 117600424 | C/T | 0.409 | 1.00 | 0.0072 (0.0012) | — | $1.95 \times 10^{-9}$ | 0.0093 (0.0061) | 0.130 | 0.0073 (0.0012) | $6.37 \times 10^{-10}$ | ++ |
| | 9 | rs263575 | 17033840 | A/G | 0.460 | 1.00 | −0.0066 (0.0012) | — | $2.31 \times 10^{-8}$ | −0.0157 (0.0061) | $9.45 \times 10^{-3}$ | −0.0069 (0.0012) | $2.23 \times 10^{-9}$ | −− |
| | 10 | rs1021363 | 106610839 | G/A | 0.642 | 1.00 | −0.007 (0.0012) | SORCS3 | $1.04 \times 10^{-8}$ | −0.031 (0.0063) | $9.34 \times 10^{-7}$ | −0.0079 (0.0012) | $5.54 \times 10^{-11}$ | −− |
| | 11 | rs10501696 | 88748162 | G/A | 0.499 | 0.99 | −0.0079 (0.0012) | GRM5 | $6.73 \times 10^{-11}$ | −0.0251 (0.0066) | $1.49 \times 10^{-4}$ | −0.0084 (0.0012) | $1.24 \times 10^{-12}$ | −− |
| | 13 | rs9530139 | 31847324 | T/C | 0.195 | 1.00 | −0.0089 (0.0015) | B3GLCT | $2.63 \times 10^{-9}$ | −0.0075 (0.0078) | 0.338 | −0.0088 (0.0015) | $1.66 \times 10^{-9}$ | −− |
| | 15 | rs28541419 | 88945878 | G/C | 0.231 | 1.00 | −0.0078 (0.0014) | — | $2.78 \times 10^{-8}$ | −0.0029 (0.0073) | 0.688 | −0.0076 (0.0014) | $3.18 \times 10^{-8}$ | −− |
| Probable MDD | 2 | rs10929355 | 15398964 | G/T | 0.456 | 1.00 | −0.0075 (0.0013) | NBAS | $5.84 \times 10^{-9}$ | −0.0078 (0.0061) | 0.199 | −0.0075 (0.0013) | $2.50 \times 10^{-9}$ | −− |
| | 7 | rs5011432 | 12268668 | C/A | 0.412 | 1.00 | 0.0073 (0.0013) | TMEM106B | $2.23 \times 10^{-8}$ | 0.022 (0.0061) | $3.15 \times 10^{-4}$ | 0.008 (0.0013) | $4.85 \times 10^{-10}$ | ++ |
| ICD-coded MDD | 7 | rs1554505 | 1983929 | A/G | 0.752 | 1.00 | 0.004 (0.0007) | MAD1L1 | $2.74 \times 10^{-9}$ | 0.017 (0.007) | 0.015 | 0.0042 (0.0007) | $7.58 \times 10^{-10}$ | ++ |

Variants were examined within the 23andMe association analysis of depression[4] to obtain their reported P-values and determine whether their effect was in the same direction as UK Biobank
The allele frequency (Freq) is for the A1 allele within UK Biobank, with the effect size (Beta) and standard error (St. err.) reported for the A1 allele within UK Biobank, 23andMe and the meta-analysis. The chromosome (Chr) and basepair position is given with regards to the GRCh37 assembly. Imputation accuracy (Info) score of UK Biobank was calculated based on the sample analysed

depression phenotypes and schizophrenia[15] ($0.29$ ($\pm 0.04$) $\leq r_g \leq 0.30$ ($\pm 0.05$), $P_{FDR} \leq 4.60 \times 10^{-9}$). Broad depression and probable MDD were significantly genetically correlated with bipolar disorder[16] ($r_g = 0.33$ ($\pm 0.07$), $P_{FDR} \leq 5.44 \times 10^{-5}$) and broad depression was genetically correlated with attention deficit hyperactivity disorder (ADHD)[17] ($r_g = 0.36$ ($\pm 0.11$), $P_{FDR} = 0.01$). No significant genetic correlations ($P_{FDR} > 0.33$) were found between autism spectrum disorder and the three depression-related phenotypes.

**Gene and region-based analyses**. We used the MAGMA[18] package to identify genes with a significant effect ($P < 2.77 \times 10^{-6}$) on each phenotype. There were 78 genes significantly associated with broad depression (Supplementary Data 6), two genes that were associated with probable MDD (Supplementary Data 7) and one gene that was associated with ICD-coded MDD (Supplementary Data 8).

We also used MAGMA to identify genomic regions, defined by recombination hotspots, with a statistically significant effect ($P < 6.02 \times 10^{-6}$) on each phenotype. There were 59 significant regions identified for broad depression, four regions identified for probable MDD, and four regions for ICD-coded MDD. Further details regarding these regions are provided in Supplementary Data 9, 10, and 11 for broad depression, probable MDD and ICD-coded MDD, respectively.

Manhattan plots of the gene and regions analysed are provided in Supplementary Figs. 7, 8, and 9 for broad depression, probable MDD, and ICD-coded MDD, respectively.

**Gene-set pathway analysis**. We conducted gene-set enrichment analysis[19,20] and identified five significant pathways for broad depression after applying correction for multiple testing; GO_EXCITATORY_SYNAPSE (beta = $0.346 \pm 0.069$, $P_{corrected} = 0.004$), GO_MECHANOSENSORY_BEHAVIOR (beta = $1.390 \pm 0.290$, $P_{corrected} = 0.009$), GO_POSTSYNAPSE (beta = $0.241 \pm 0.050$, $P_{corrected} = 0.009$), GO_NEURON_SPINE (beta = $0.376 \pm 0.085$, $P_{corrected} = 0.035$) and GO_DENDRITE (beta = $0.195 \pm 0.045$, $P_{corrected} = 0.041$) (Table 2). The genes within each of these significant pathways is provided in Supplementary Data 12 and the gene overlap between each pathway are provided in Supplementary Table 6. No pathways were associated ($P > 0.05$) with probable MDD or ICD-coded MDD after multiple testing correction.

**Tissue enrichment analysis**. The summary statistics from the three phenotypes were examined for enrichment in 209 different tissues using DEPICT[21]. None of the tissues showed significant enrichment across the three depression-related phenotypes, after applying the default false discovery rate correction (Supplementary Data 13). However, within broad depression the tissues that showed greatest enrichment were those involved in the brain and nervous system.

**eQTL identification**. Of the 17 variants associated with depression, 7 were identified as eQTLs (Supplementary Data 14); of these, 4 variants were found to be eQTL for brain expressed genes. rs6699744 is associated with the expression of *RPL31P12* in

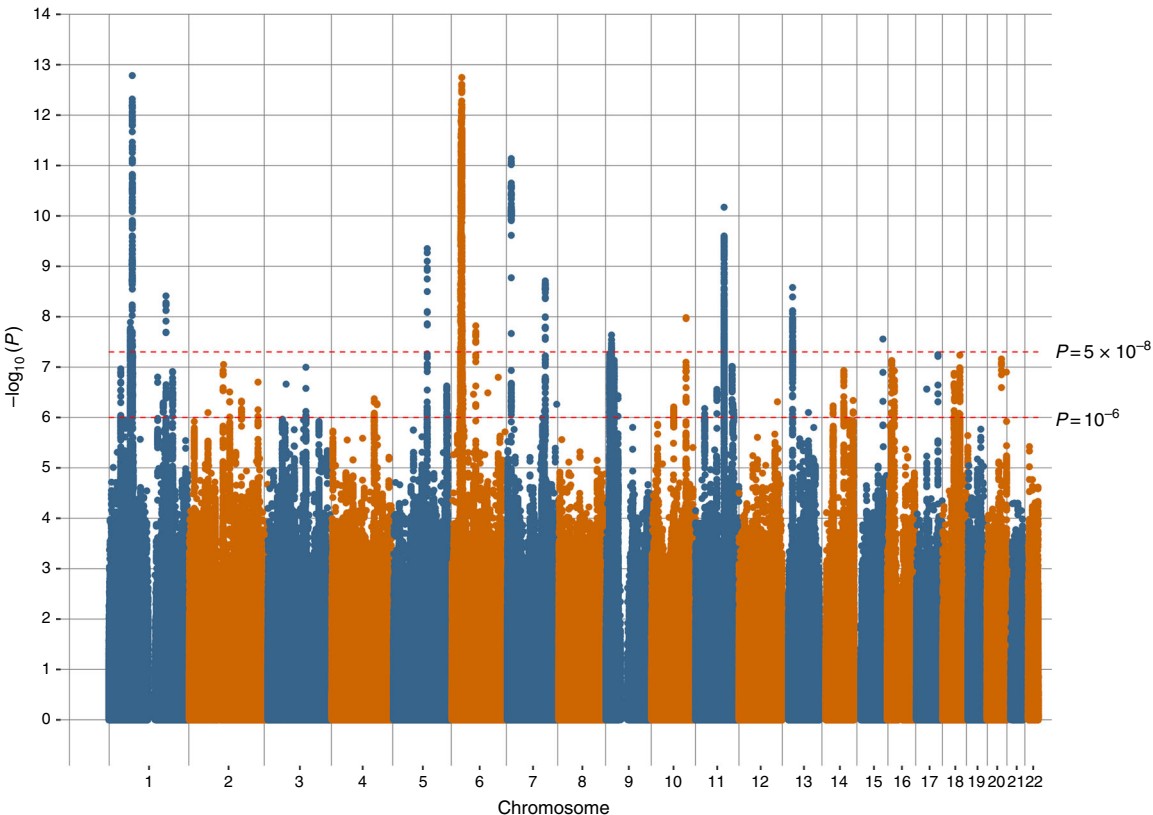

**Fig. 1** Manhattan plot of the observed $-\log_{10}$ $P$-values of each variant for an association with broad depression ($n = 322,580$) in the UK Biobank cohort. Variants are positioned according to the GRCh37 assembly

the cerebellum, and rs9530139 is associated with expression of *B3GALTL* in the cortex. rs40465 and rs68141011 are broad eQTLs affecting the expression of a number of zinc finger protein encoding genes across various brain tissues, including *ZNF391, ZNF204P, ZNF192P1, ZSCAN31* and *ZSCAN23*. None of these brain expressed genes overlapped with the genes within the five significant gene-pathways highlighted previously.

## Discussion

This study describes a large analysis of depression-related phenotypes using a single population-based cohort. Up to 322,580 individuals from the UK Biobank cohort were used to test the effect of 7,666,894 genetic variants on three depression phenotypes. A total of 17 independent genome-wide significant ($P < 5 \times 10^{-8}$) variants were identified across the three phenotypes. Replication was sought for these 17 variants within the 23andMe cohort[4]. We found that 16 of these variants had an effect in the same direction as UK Biobank, with seven of the variants in the replication cohort being significant after correction for multiple testing ($P < 0.0029$). The broadest definition of the phenotype, broad depression, provided the greatest number of individuals for analysis and also the largest number of significant hits (14 independent variants). The probable MDD phenotype was obtained using the approach of Smith, et al.[10], using responses to touchscreen items that were only administered in the last two years of UK Biobank recruitment. Thus, the probable MDD phenotype was available for approximately one-third of the sample, and yielded two independent genome-wide significant variants. The strictest phenotype, ICD-coded MDD (using linked hospital admission records for a primary or secondary diagnosis of ICD 9/10 MDD), had one independent significant variant.

Strong positive genetic correlations were found between the three UK Biobank phenotypes and a mega-analysis of MDD, based on an anchor set of clinically defined cases. Interestingly, the highest genetic correlation with the clinically defined MDD phenotype was for the broad depression phenotype. The broad depression phenotype was genetically correlated with the greatest number of other traits, including schizophrenia, bipolar disorder and ADHD, with these correlations in general alignment with those reported by Major Depressive Disorder Working Group of the Psychiatric Genomics Consortium., et al.[3]. The very high genetic correlations between the three phenotypes for depression within UK Biobank is not surprising as there is overlap between individuals classified as cases. However, we also showed significant differences between the underlying genetic architectures of the three phenotypes, and these differences may be informative for defining depression-related phenotypes.

Where each phenotype differed most, was in its tractability for discovering specific genetic loci with UK Biobank. Despite the majority of the cross-phenotype effect sizes being in the same direction (32 out of 34), only half demonstrated a nominally significant association ($P < 0.05$). This variability in the variants underlying each phenotype potentially suggests that depression phenotypes may differ in their tractability for genetic studies. The number of cases for probable MDD ($n = 30,603$) and ICD-coded MDD ($n = 8,276$) was considerably lower than for broad depression ($n = 113,769$). The power to detect significant effects for probable MDD and ICD-coded MDD could be limited, because at those sample sizes, no variants have yet achieved replicable associations for depression. Whilst false positive associations could also explain non-overlap between the findings for the three UK Biobank depression phenotypes, replication of our findings in an independent cohort suggests that this is not the case.

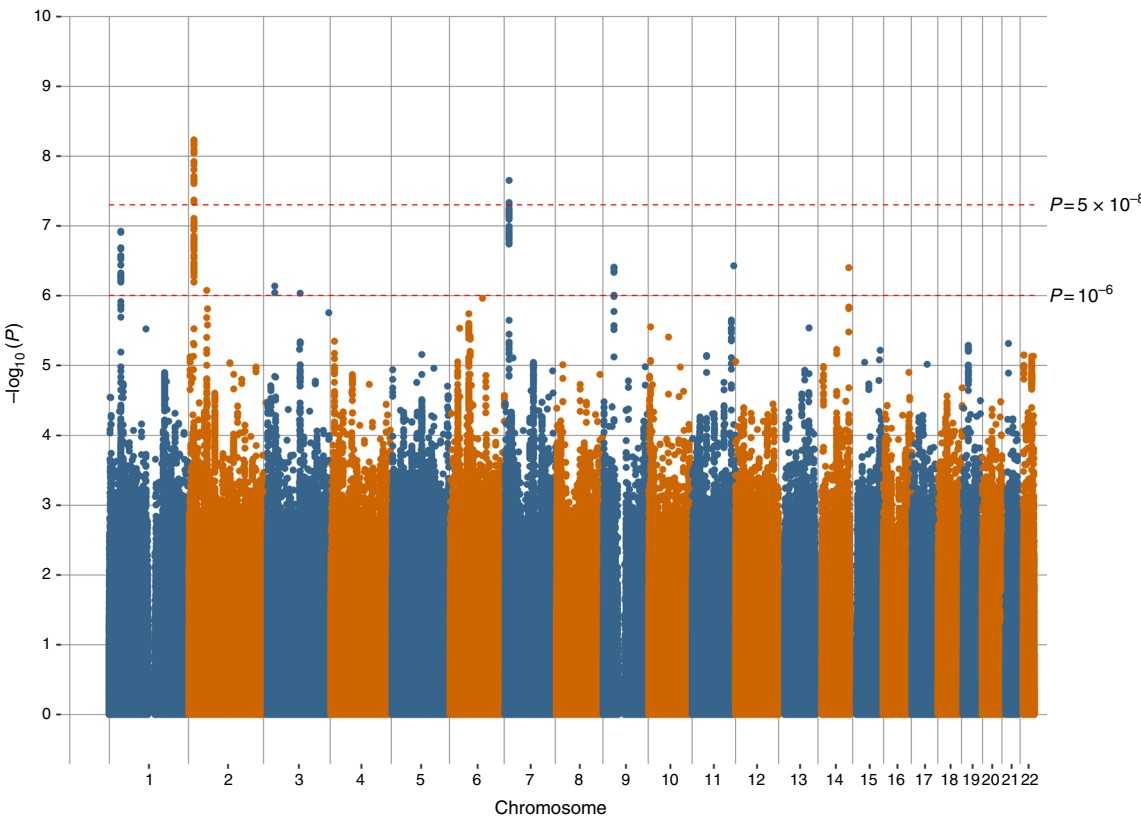

**Fig. 2** Manhattan plot of the observed $-\log_{10}$ P-values of each variant for an association with probable major depressive disorder ($n = 174{,}519$) in the UK Biobank cohort. Variants are positioned according to the GRCh37 assembly

There were a number of genes identified as associated with depression, as well as genes that overlapped with the associated variants. For brevity of this discussion, these genes and their putative effects and biological pathways are covered in greater depth in Supplementary Note 1. The genes that are potentially of interest for the study of depression include the Neural Growth Regulator 1 (*NEGR1*), the glutamate metabotropic receptor 5 (*GRM5*), the glutamate ionotropic receptor kainate type subunit 3 (*GRIK3*) and the transmembrane protein 106B (*TMEM106B*) protein coding genes.

Five gene sets were significantly enriched in broad depression. Of these, four were associated with cellular components (where the genes are active), and one was associated with a biological process. The cellular components were all associated with parts of the nervous system (excitatory synapse, neuron spine, postsynapse and dendrite) and demonstrates that genes that are active in these components could be attributing to depression. However, there was considerable overlap between the genes that were involved in the cellular components. Excitatory synapses are the site of release for excitatory neurotransmitters and have been previously associated with depression- related phenotypes[22,23], with the most common excitatory neurotransmitter, glutamate, also demonstrating associations with depression[24,25]. The detection of this pathway, using a genome-wide association analysis approach, allows the specific genes driving this association to be identified, with sortilin-related VPS10 domain containing receptor 3 (*SORCS3*), *GRM5*, dopamine receptor D2 (*DRD2*) and calcium binding protein 1 (*CABP1*) the most significant genes for broad depression in this pathway. Neuron spines (or dendritic spines) are extensions from dendrites that act as a primary site for excitatory transmission in the brain. Imbalances of excitation and inhibition have been previously associated with other mental

disorders such as schizophrenia, Tourette's syndrome and autism spectrum disorder[26]. These results indicate that the role of excitatory synapses in the pathology of depression should be further investigated.

Mechanosensory behaviour refers to behaviour that is prompted from a mechanical stimulus (e.g. physical contact with an object). This indicates that depressive individuals may have a differing behavioural response to mechanosensory stimuli, with higher pain sensitivity and lower pain pressure thresholds found in depression cases[27]. The identification of the genes involved in each of the pathways reported may ultimately be informative of the underlying biological processes for each phenotype.

We derived three definitions of depression in the UK Biobank cohort. The broad depression phenotype provided the greatest number of cases and was based on self-reported help-seeking behaviour via a general practitioner or psychiatrist, and therefore is likely to also capture other personality disorders. To examine potential overlap with generalised anxiety disorder we used LDSR[13] to calculated genetic correlations with our three depression-related phenotypes and anxiety factors score from Otowa, et al.[28]. There was no clear evidence for additional enrichment for anxiety disorder in the broad depression phenotype ($r_g = 0.052 \pm 0.11$) compared to probable MDD ($r_g = 0.060 \pm 0.17$) or ICD-coded MDD ($r_g = 0.047 \pm 0.17$). The probable MDD supplemented help-seeking behaviour with additional information relating to low mood and/or anhedonia over a two-week period and the ICD-coded MDD phenotype was based on a hospital-treated and healthcare-professional-coded diagnosis of MDD. The ICD-coded MDD phenotype provided substantially fewer cases, many of which were secondary, or potentially incidental, to medical diagnoses.

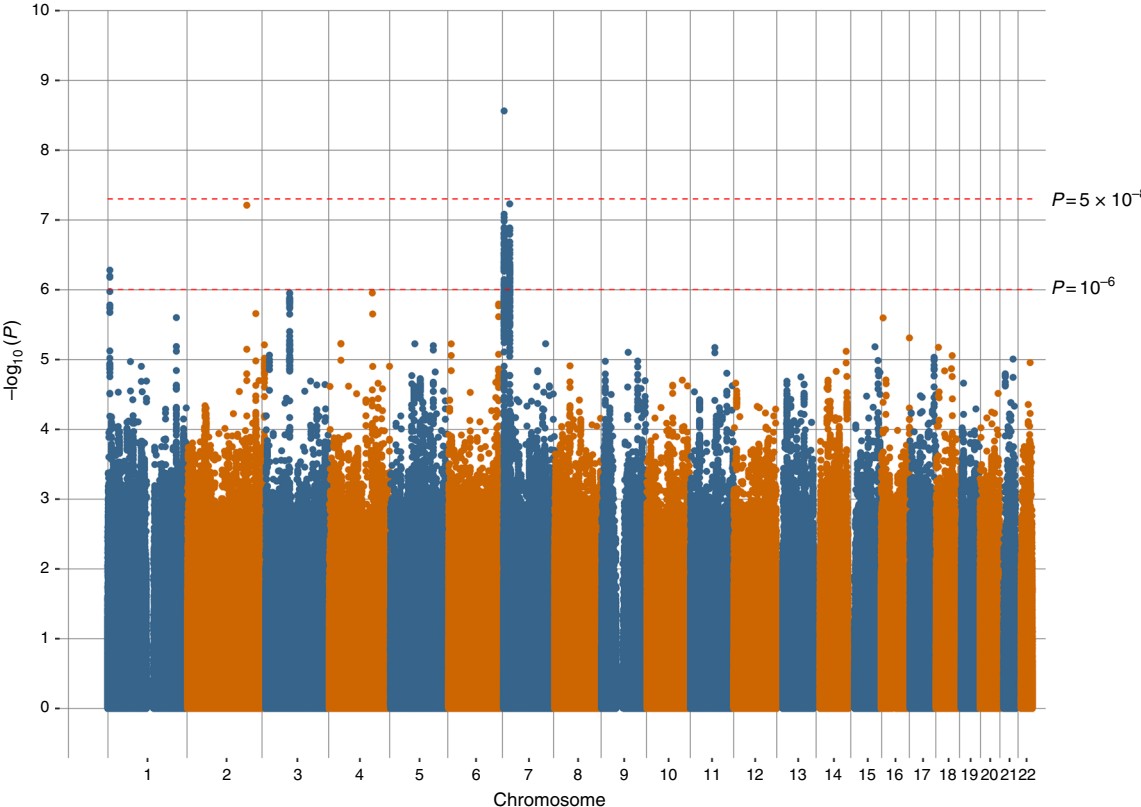

**Fig. 3** Manhattan plot of the observed −log10 P-values of each variant for an association with International Classification of Diseases-coded major depressive disorder (n = 217,584) in the UK Biobank cohort. Variants are positioned according to the GRCh37 assembly

Although useful for genetic studies, each depression phenotype has limitations. None are based on a formal structured diagnostic assessment (such as the Structured Clinical Interview for DSM Axis 1 Disorders interview), and both the broad depression and probable MDD phenotypes are based on self-reported information, which can be subject to recall biases. Broad depression is also likely to be endorsed by a wider range of individuals than traditional depression definitions, including those with internalising disorders other than depression and those with depressive symptoms that would not meet diagnostic criteria for MDD. The ICD-coded MDD phenotype is based on hospital admission records, which can sometimes be incomplete as not all participants would have a hospital admission leading to a diagnosis of MDD. This phenotype provided just one genome-wide significant variant which may represent the extent of the detectable signal available for a clinically MDD diagnosis in a population of the size studied.

However, in alignment with Hyde et al.[4], we found that self-reported measures of depression were highly genetically correlated with those obtained from the clinically-diagnosed depression phenotype[6]. Therefore, the analysis of large cohorts with self-reported depression appears to offer greater power for detecting genetic effects than smaller studies with a more robustly defined clinical phenotype. Depression is, however, a heterogeneous condition[29,30] and to gain greater insight into its subtypes, larger studies are likely to be necessary to improve diagnosis, treatment and patient outcomes. This approach might only be possible with relatively easy-to-recruit, large, population-based self-declared cohorts, supplemented where feasible with additional clinical and biological data.

In a large genome-wide association study of three depression-related phenotypes in UK Biobank, we identified 17 risk variants.

In a replication sample, consistency of direction of effect was seen with seven variants formally replicated. Further analysis of our results identified genetic correlations with a number of other psychiatric disorders and implicating perturbations of excitatory neurotransmission in depression. Our results suggest that a broad depression phenotype, which potentially overlaps with other personality and psychiatric traits, may provide a more tractable target for future genetic studies, allowing the inclusion of many more samples.

## Methods

**Study population.** The UK Biobank cohort is a population-based cohort consisting of 501,726 individuals recruited at 23 centres across the United Kingdom. Genotypic data was available for 488,380 individuals and was imputed with IMPUTE4 using the HRC reference panel[31] to identify ~39 M variants for 487,409 individuals[32]. We excluded 79,990 individuals that were outliers based on heterozygosity, had a variant call rate <98%, or were not recorded as "white British". We excluded a further 131,790 related individuals based on a shared relatedness of up to the third degree using kinship coefficients (>0.044) calculated using the KING tool-set[33]; we then subsequently added back in one member of each group of related individuals by creating a genomic relationship matrix and selected individuals with a genetic relatedness less than 0.025 with any other participant (n = 55,745). We removed variants with a call rate < 98%, a minor allele frequency <0.01, deviation from Hardy–Weinberg equilibrium ($P < 10^{-6}$) or an imputation accuracy (Info) score < 0.1, leaving a total of 7,666,894 variants for 331,374 individuals.

Extensive phenotypic data were collected for UK Biobank participants using health records, biological sampling, physical measures and touchscreen tests and questionnaires. We used three definitions of depression in the UK Biobank sample, which are explained in greater depth in Supplementary Note 2 and are described below. For the three UK Biobank depression phenotypes, we excluded participants who were identified with bipolar disorder, schizophrenia or personality disorder using self-declared data, touchscreen responses (per Smith, et al.[10]) or ICD codes from hospital admission records; and participants who reported having a prescription for an antipsychotic medication during a verbal interview. Further exclusions were applied to all control individuals if they had a diagnosis of a depressive mood disorder from hospital admission records, had reported having a

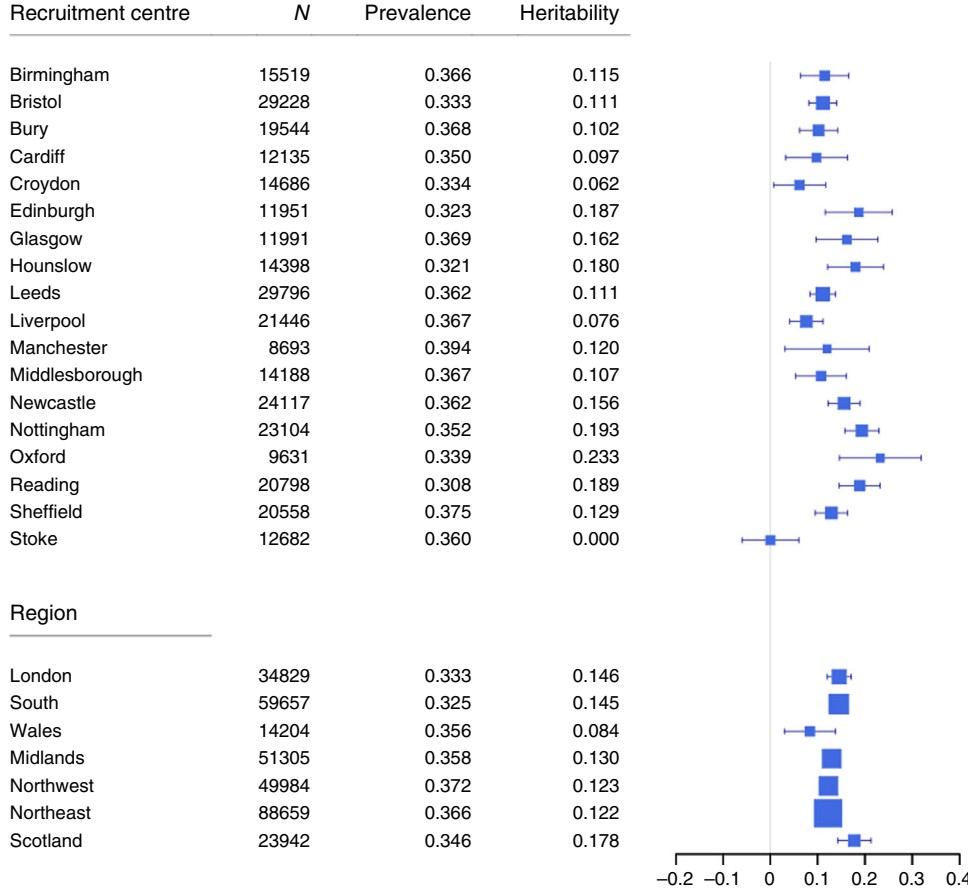

| Recruitment centre | N | Prevalence | Heritability |
|---|---|---|---|
| Birmingham | 15519 | 0.366 | 0.115 |
| Bristol | 29228 | 0.333 | 0.111 |
| Bury | 19544 | 0.368 | 0.102 |
| Cardiff | 12135 | 0.350 | 0.097 |
| Croydon | 14686 | 0.334 | 0.062 |
| Edinburgh | 11951 | 0.323 | 0.187 |
| Glasgow | 11991 | 0.369 | 0.162 |
| Hounslow | 14398 | 0.321 | 0.180 |
| Leeds | 29796 | 0.362 | 0.111 |
| Liverpool | 21446 | 0.367 | 0.076 |
| Manchester | 8693 | 0.394 | 0.120 |
| Middlesborough | 14188 | 0.367 | 0.107 |
| Newcastle | 24117 | 0.362 | 0.156 |
| Nottingham | 23104 | 0.352 | 0.193 |
| Oxford | 9631 | 0.339 | 0.233 |
| Reading | 20798 | 0.308 | 0.189 |
| Sheffield | 20558 | 0.375 | 0.129 |
| Stoke | 12682 | 0.360 | 0.000 |

| Region | N | Prevalence | Heritability |
|---|---|---|---|
| London | 34829 | 0.333 | 0.146 |
| South | 59657 | 0.325 | 0.145 |
| Wales | 14204 | 0.356 | 0.084 |
| Midlands | 51305 | 0.358 | 0.130 |
| Northwest | 49984 | 0.372 | 0.123 |
| Northeast | 88659 | 0.366 | 0.122 |
| Scotland | 23942 | 0.346 | 0.178 |

**Fig. 4** Forest plot of the estimated SNP-based heritability of broad depression by recruitment centre and region. Heritabilities are provided where there was power of at least 30% to detect a heritability >0 with a trait heritability of 9%, a type I error of 0.05, a trait prevalence of 0.3527 and a variance of the SNP-derived genetic relationships of $2 \times 10^{-5}$

prescription for antidepressants or had self-reported depression (see Supplementary Note 2 for full phenotype criteria).

This research has been conducted using the UK Biobank Resource—application number 4844. The UK Biobank study was conducted under generic approval from the NHS National Research Ethics Service (approval letter dated 17th June 2011, Ref 11/NW/0382). All participants gave full informed written consent.

**Broad depression phenotype**. The broadest phenotype (broad depression) was defined using self-reported help-seeking behaviour for mental health difficulties. Case and control status was determined by the touchscreen response to either of two questions: "Have you ever seen a general practitioner (GP) for nerves, anxiety, tension or depression?" (UK Biobank field: 2090) or "Have you ever seen a psychiatrist for nerves, anxiety, tension or depression?" (UK Biobank field 2010). Caseness for broad depression was determined by answering "Yes" to either question at either the initial assessment visit, at any repeat assessment visit, or if there was a primary or secondary diagnosis of a depressive mood disorder from linked hospital admission records (UK Biobank fields: 41202 and 41204; ICD codes: F32—Single Episode Depression, F33—Recurrent Depression, F34—Persistent mood disorders, F38—Other mood disorders and F39—Unspecified mood disorders). The remaining respondents were classed as controls if they provided "No" responses to both questions during all assessments that they participated in. This provided a total of 113,769 cases and 208,811 controls ($n_{total} = 322,580$, prevalence = 35.27%) for the broad depression phenotype. Individuals classed as cases for the broad depression phenotype potentially included individuals seeking treatment for personality disorders.

**Probable MDD phenotype**. The second depression phenotype (probable MDD) was derived from touchscreen responses to questions about the presence and duration of low mood and anhedonia, following the definitions from Smith et al.[10], whereby the participant had indicated that they were "depressed/down for a whole week (UK Biobank field: 4598); plus at least 2 weeks duration (UK Biobank field: 4609); plus ever seen a GP or psychiatrist for nerves, anxiety or depression" (UK Biobank fields: 2090 and 2010), or "ever anhedonia for a whole week (UK Biobank field: 4631); plus at least 2 weeks duration (UK Biobank field: 5375); plus ever seen

a GP or psychiatrist for nerves, anxiety, or depression" (UK Biobank fields: 2090 and 2010). Cases for the probable MDD definition were supplemented by diagnoses of depressive mood disorder from linked hospital admission records (UK Biobank fields: 41202 and 41204) as per the broad depression phenotype. There were a total of 30,603 cases and 143,916 controls ($n_{total} = 174,519$, prevalence = 17.54%) for the probable MDD phenotype.

There were 66,176 individuals that were classed as controls for probable MDD, who were classed as cases for broad depression. This was due to stricter conditions for classification as a case for probable MDD compared to the broad depression phenotype, and potentially removes individuals seeking treatment for personality disorders.

**ICD-coded MDD phenotype**. The ICD-coded MDD phenotype was derived from linked hospital admission records (UK Biobank fields: 41202 and 41204). Participants were classified as cases if they had either an ICD-9/10 primary or secondary diagnosis for a depressive unipolar mood disorder (ICD codes: F32—Single Episode Depression, F33—Recurrent Depression, F34—Persistent mood disorders, F38—Other mood disorders and F39—Unspecified mood disorders). ICD-coded MDD controls were participants who had linked hospital records, but who did not have any diagnosis of a mood disorder and were not probable MDD cases. There were 8,276 cases and 209,308 controls ($n_{total} = 217,584$, prevalence = 3.80%) for the ICD-coded MDD phenotype.

There were no individuals classed as a case for ICD-coded MDD who were classed as a control for either the broad depression or probable MDD phenotypes. There were no individuals classed as controls in ICD-coded MDD who were classed as a case for probable MDD. However, there were 53,491 control individuals for ICD-coded MDD who were classed as a control for broad depression, which reflects the potentially looser definition stipulated under the broad depression phenotype. Of the 8,276 individuals classed as a case using the ICD-coded MDD definition, 7,471 (90.3%) also reported seeing either a GP or psychiatrist for nerves, anxiety, or depression (UK Biobank fields: 2090 and 2010), or had been depressed/down or suffered from anhedonia for a whole week (UK Biobank field: 4598).

A cross tabulation of case and control status between each pair of phenotypes is provided in Supplementary Table 7.

**Table 2 Pathways with a significant effect ($P_{corrected} < 0.05$) on broad depression following multiple testing correction identified through gene-set enrichment analysis**

| Phenotype | Pathway | Number of genes | Beta (St. Err.) | P-value | P_Corrected |
|---|---|---|---|---|---|
| Broad depression | GO_EXCITATORY_SYNAPSE | 182 | 0.346 (0.069) | $2.38 \times 10^{-7}$ | 0.004 |
| | GO_MECHANOSENSORY_BEHAVIOR | 11 | 1.390 (0.290) | $7.99 \times 10^{-7}$ | 0.009 |
| | GO_POSTSYNAPSE | 352 | 0.241 (0.050) | $8.26 \times 10^{-7}$ | 0.009 |
| | GO_NEURON_SPINE | 114 | 0.376 (0.085) | $4.89 \times 10^{-6}$ | 0.035 |
| | GO_DENDRITE | 423 | 0.195 (0.045) | $6.06 \times 10^{-6}$ | 0.041 |

**Association analysis**. We performed a linear association test to assess the effect of each variant using BGENIE v1.1[32]:

$$\mathbf{y} = \mathbf{X\beta} + \boldsymbol{\varepsilon}_1$$
$$\hat{\mathbf{y}} = \mathbf{X\beta}$$
$$(\mathbf{y} - \hat{\mathbf{y}}) = \mathbf{G}b + \boldsymbol{\varepsilon}_2$$

where $\mathbf{y}$ was the vector of binary observations for each phenotype (controls coded as 0 and cases coded as 1). $\boldsymbol{\beta}$ was the matrix of fixed effects, including sex, age, genotyping array, and 8 principal components. $\mathbf{X}$ was the corresponding incidence matrices. $(\mathbf{y} - \hat{\mathbf{y}})$ was a vector of phenotypes residualized on the fixed effect predictors, $\mathbf{G}$ was a vector of expected genotype counts of the effect allele (dosages), b was the effect of the genotype on residualized phenotypes, and $\boldsymbol{\varepsilon}_1$ and $\boldsymbol{\varepsilon}_2$ were vectors of normally distributed errors.

Genome-wide statistical significance was determined by the conventional threshold of $P < 5 \times 10^{-8}$. To determine significant variants that were independent, the clump command in Plink 1.90b4[11] was applied using --clump-p1 1e-4 --clump-p2 1e-4 --clump-r2 0.1 --clump-kb 3000, mirroring the approach of Major Depressive Disorder Working Group of the Psychiatric Genomics Consortium., et al.[3]. Therefore, variants that were within 3 Mb of each other and shared a linkage disequilibrium greater than 0.1 were clumped together, and only the most significant variant was reported.

Due to the complexity of major histocompatibility complex (MHC) region, an approach similar to that of The Schizophrenia Psychiatric Genome-Wide Association Study Consortium[34] was taken, and only the most significant variant across that region was reported. To obtain odds ratios for those variants associated with depression, a logistic regression was conducted in Plink 1.90b4[11] using the same covariates as the linear model above. Regional visualisation plots were produced using LocusZoom[35]. Linkage Disequilibrium Score regression (LDSR)[13] was used to determine whether there was elevation of the polygenic signal due to population stratification, by examining the intercept for evidence of significant deviation ($\pm 1.96$ standard error) from 1. The genomic inflation factor ($\lambda_{GC}$) was also reported for each phenotype.

**Replication cohort and meta-analysis**. We sought to replicate those variants within UK Biobank that were identified as significantly associated ($P < 5 \times 10^{-8}$) with depression. To conduct the replication, we used the associated analysis results from the discovery sample from the 23andMe cohort[4] ($n = 307,354$, cases $= 75,607$ and controls $= 231,747$). We firstly examined if the effect of the A1 allele was in the same direction across both UK Biobank and 23andMe. Secondly, we determined whether the variant within 23andMe was significant for the 17 variants ($\alpha = 0.05 / 17$; $P < 0.0029$). Additionally, we used Metal[36] to conduct an inverse variance-weighted meta-analysis of these 17 significant variants in UK Biobank and 23andMe.

**Heritability of depression**. We used GCTA-GREML[12] to estimate the SNP-based $h^2$ for each depression phenotype within each recruitment centre and each region. The specified population prevalence was matched to that observed across the whole cohort for each phenotype to allow transformation from the observed to the underlying liability scale. The recruitment centres within each region is provided in Supplementary Table 4. LDSR[13] was used to provide a whole sample SNP-based estimate of the $h^2$ on liability scale of the phenotypes using the whole-genome summary statistics obtained by the association analyses. The summary statistics for each phenotype were filtered using the default file, w_hm3.snplist, with the default LD Scores computed using 1000 Genomes European data (eur_w_ld_chr) used as a reference.

**Genetic correlations with depression**. To calculate the genetic correlations between the three depression-related phenotypes in UK Biobank LDSR[13] was used. We also tested that hypothesis that the phenotypes were perfectly correlated with one another ($r_g = 1$):

$$P - \text{value} = 2 \times \text{pnorm}\left(-\text{abs}\left(\frac{1 - r_g}{\text{standard error}}\right)\right)$$

Genetic correlations were also calculated between each of the three phenotypes and 235 other behavioural and disease related traits using LD Hub[14]. P-values were false discovery rate (FDR) adjusted using the Benjamini and Hochberg[37] approach.

**Gene- and region-based analyses**. Two downstream analyses of the results were conducted using MAGMA[18] (Multi-marker Analysis of GenoMic Annotation) by applying a multiple regression model, which incorporates linkage disequilibrium between markers and accounts for multi-marker effects, to the results of our association analyses. Estimates of the linkage disequilibrium between markers were obtained from the European populations sequenced as part of the 1,000 Genomes Project (phase 1, release 3)[38]. In the first downstream analysis, a gene-based analysis was performed for each phenotype using the results from our GWAS. The NCBI 37.3 build was used to determine the genetic variants that were ascribed to each gene. A total of 18,033 genes were assessed for an association with each depression phenotype with Bonferroni correction used to determine significance ($\alpha = 0.05 / 18,033$; $P < 2.77 \times 10^{-6}$).

In the second downstream analysis, a region-based analysis was performed for each phenotype. To determine the regions, haplotype blocks identified by recombination hotspots were used as described by Shirali, et al.[39] and implemented in an analysis of MDD by Zeng, et al.[40] for detecting causal regions. Block boundaries were defined by hotspots of at least 30 cM per Mb based on a European subset of the 1,000 genome project recombination rates. This resulted in a total of 8,308 regions being analysed using the European panel of the 1,000 Genomes data (phase 1, release 3) as a reference panel to account for linkage disequilibrium. A genome-wide significance threshold for region-based associations was calculated using the Bonferroni correction method ($\alpha = 0.05 / 8,308$; $P < 6.02 \times 10^{-6}$).

**Gene-set pathway analysis**. We used the results obtained from our gene-based analysis to conduct a further gene-set pathway analysis to test for gene enrichment within 5,917 biological classes using MAGMA[18]. The gene-set pathways were obtained from the gene-annotation files provided by the Gene Ontology (GO) Consortium (http://geneontology.org/)[19], which are derived from the Molecular Signatures Database (MSigDB) v5.2[20]. To correct for multiple testing, the default of 10,000 permutations was applied. The estimate of the effect size (beta) reflects the difference in association between genes in the gene set and genes outside the gene set from fitting a regression model to the data.

**Tissue enrichment analysis**. The DEPICT package[21] was used to investigate whether specific tissues were enriched using the genome-wide association study results obtained for each phenotype. The recommended P-value threshold of $<10^{-5}$ was applied to the variants used to assess enrichment. DEPICT incorporates data from 37,427 human gene expression microarrays to determine whether genes in associated loci for each phenotype were significantly expressed in any of 209 tissue/cell type annotations.

**eQTL identification**. The online GTEx portal (https://www.gtexportal.org/home/) was used to determine whether any of the genome-wide significant variants for each phenotype were eQTL[9].

**Code availability**. All code used to conduct the research detailed in this manuscript is available on request from the corresponding author.

**Data availability**. The summary statistics relating to each of the three depression phenotypes assessed in this study are available on the Edinburgh DataShare website, via the following link: https://doi.org/10.7488/ds/2314

The raw genetic and phenotypic data that support the findings of this study are available from UK Biobank but restrictions apply to the availability of these data, which were used under license for the current study, and so are not publicly available. Data are, however, available from the authors upon reasonable request and with permission of UK Biobank (http://www.ukbiobank.ac.uk/).

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

## Acknowledgements

The authors acknowledge the help, advice and support from all members of the UK Biobank Psychiatric Genetics Group. We are grateful to the research participants and employees of UK Biobank and 23andMe. A.M.McI, and I.J.D. acknowledge support from the Wellcome Trust (Wellcome Trust Strategic Award "STratifying Resilience and Depression Longitudinally" (STRADL) Reference 104036/Z/14/Z and the Sackler Foundation. I.J.D. is supported by the Centre for Cognitive Ageing and Cognitive Epidemiology, which is funded by the Medical Research Council and the Biotechnology and Biological Sciences Research Council (MR/K026992/1). This investigation represents independent research part-funded by the National Institute for Health Research (NIHR) Biomedical Research Centre at South London and Maudsley NHS Foundation Trust and King's College London. The views expressed are those of the authors and not necessarily those of the NHS, the NIHR or the Department of Health. DJS supported by Lister Institute Prize Fellowship 2016–2021.

## Author contributions

D.M.H., D.J.S., G.B., C.M.L. and A.M.McI. conceived the research project. M.J.A., J.R.I.C., J.W., D.J.S., G.B. and A.M.McI. determined the variables that formed the depression phenotypes within UK Biobank. D.M.H., M.J.A., J.R.I.C., R.E.M., S.P.H. and G.D. applied quality control to the UK Biobank data. M.J.A. ran the association analysis in UK Biobank. 23andMe R.T. provided the summary statistics from the Hyde, et al.[4] analysis. D.M.H ran the h[2] analysis, calculated the genetic correlations and ran the tissue enrichment analysis. M.S. ran the MAGMA analysis with J.W., D.M.H., T-K.C., J.G., E.M.W., C.A., X.S. and M.C.B. examining the genes and gene sets identified by the MAGMA analysis. T-K.C., conducted the eQTL identification analysis. I.J.D., C.S.H., D.J.S., P.F.S., G.B., C.M.L. and A.M.McI. provided expertise of association study methodology and statistical analysis. D.M.H. oversaw the research project and serves as the primary contact for all communication. All authors commented on the manuscript.

## Additional information

**Competing interests:** I.J.D. is a participant in UK Biobank. The remaining authors declare no competing interests.

## 23andMe Research Team

Michelle Agee[11], Babak Alipanahi[11], Adam Auton[11], Robert K. Bell[11], Katarzyna Bryc[11], Sarah L. Elson[11], Pierre Fontanillas[11], Nicholas A. Furlotte[11], David A. Hinds[11], Karen E. Huber[11], Aaron Kleinman[11], Nadia K. Litterman[11], Jennifer C. McCreight[11], Matthew H. McIntyre[11], Joanna L. Mountain[11], Elizabeth S. Noblin[11], Carrie A.M. Northover[11], Steven J. Pitts[11], J. Fah Sathirapongsasuti[11], Olga V. Sazonova[11], Janie F. Shelton[11], Suyash Shringarpure[11], Chao Tian[11], Joyce Y. Tung[11], Vladimir Vacic[11] & Catherine H. Wilson[11]

[11]23andMe, Inc., Mountain View, CA 94041, USA

