## [Peer Review File · Nature Communications]

Reviewer #1 (Remarks to the Author):

In this manuscript, the authors describe the results of a GWAS for 3 depression-related phenotypes from the UK-Biobank data. They report 17 novel hits associated with these phenotypes. Gene-based and pathway analyses highlight the relevance of synaptic mechanisms in these associations. The association data show some correlation with previous studies investigating either clinically defined major depression or a personality trait identified as a risk factor for depression, neuroticism.

The strength of the data is the large sample size with uniform phenotyping and genotyping and is currently the largest reported sample for association with depression-related phenotypes (N cases).

Overall, the analyses are very solid and convincing. However, there would be possibilities to improve the relevance of the paper for the field.

- 1) the authors only compare their data to the data of Hyde et al., in which rather similar questions were used to define depression in some what smaller, but still comparable data set (23andme), for single eQTL variants. A more comprehensive comparison to this dataset would be of great interest and could also be used for more formal replication. Currently, no attempt for direct replication has been shown. How well would the data replicate in the 23andme dataset, but also the old PGC dataset with clinical phenotyping? This would also allow to better estimate what the broad depression phenotype means with respect to other phenotyping that has been used before.
- 2) The genetic correlation of 3 phenotypes seems less surprising, given the fact that overlapping phenotypes have been use, so that everybody with the ICD10 diagnosis would also be a case in the 2 other samples. At least this is how I would read the methods. The authors may want to comment on this in the paper.
- 3) The authors should give a better rationale for why additional gene-based and region-based analyses were run. What is the added information above the GWAS data?
- 4) The authors explain the lack of overlap of associations across the 3 correlated phenotypes with potential biological differences in each of these phenotypes. This needs to elaborated, given the overlap of cases. False positive associations could also be a reason for non-overlap. In the two more strict phenotypes, the N of cases drops to 30k and 8k, respectively, sample sizes for which so far no consistent association with depression could be found.
- 5) It is not clear what figure 3 really adds to the story other than showing that genes in a pathway are correlated. Please explain in the text.
- 6) The discussion reads mostly like a laundry list of genes and their prior associations with other phenotypes. A bigger picture discussion of depression phenotyping, and more detail on what does the association with the broad phenotype could mean for clinical depression would be helpful. This would be especially interesting in the light of replications in other cohort with different levels of phenotyping (self report vs. clinical).

Overall an interesting paper that could be strengthened by additional analyses and a more thoughtful discussion. Especially an attempt for replication would increase the value of the paper.

Reviewer #2 (Remarks to the Author):

The authors present GWAS of three depression phenotypes in the latest UK Biobank data. They report a number of associations, mostly with a broad self-reported phenotype of seeking care for depression or related conditions. They also present gene-based, pathway, and eQTL analyses, as well as some analyses of genetic correlations. At a high level the results are consistent with past work, that in this domain broad phenotype definitions with large sample sizes have been more successful for genetic discovery than smaller studies with more rigorous phenotyping.

The discussion mentions that several of the reported findings agree with previous studies ([3] and [4]), however, as a whole, the results are presented without replication, and it is hard to assess what is new. Can the authors look up all the index SNPs in any previous study? (maybe inconvenient for [3] since it hasn't been published yet). Also it might be useful to tabulate the GWAS results for each index SNP across the three different phenotypes: it seems worth asking how effect sizes compare for the strict vs broad phenotypes.

60-63: I might also mention Okbay et al 2016 as part of the spectrum of depression phenotypes that have been recently studied.

84: could we also see 2x2 cross tabulation of case/control status for each pair of UK Biobank phenotypes?

113-124: how can I tell whether these differences are meaningful? Can a test of heterogeneity be done, or a test for regions that differ from the overall average?

137: I might note that 11 regions are in the MHC.

Fig. 3: [editorial digression: I'm not a fan of figures whose interpretation requires me to go read other papers; even after skimming the GeneMania papers I'm not sure I could say what's going on here] What is the diagonal shading? Has the topology been optimized to pull together nodes that were associated with depression? What does "The size of each node is determined by the effect size of the gene coding region within the pathway based on previous studies" mean? (what previous studies?) Why do so many of the nodes have apparently the same size? Are there settings I should know about that affect the results? Why is this figure so nicely spaced out when the examples in the GeneMania paper are all clumpy? Why is there a monotonic progression of circle sizes along the outer ring?

184-188: the p values for the genetic correlations don't appear in Supp. Table 8 but maybe they should. Also I wonder if the more interesting test is whether these phenotypes are related (i.e. $r_g > 0$) or whether they are different (i.e. $r_g < 1$).

302: according to [60] there are ~39 million HRC markers, not 19 million? So about 80% have been excluded from analysis, I guess mostly due to the 1% MAF filter? Why was the 1% filter selected? (this excludes many variants with good information scores)

307-310: how were call rate and Hardy-Weinberg equilibrium (HWE) filters applied to imputed genotype probabilities or dosages (or am I misinterpreting)? For a sample size of >300K, is a HWE threshold of $1e-6$ rather severe? What proportion of variants are excluded by the HWE filter?

353: association tests were performed using linear regression on binary phenotypes after regressing out the covariates. Is linear regression a good solution for this problem? (I suppose it is probably safe for common variants) I might refit logistic regression models, including covariates, for the index SNPs in Table 2, at the least because that would give a more familiar effect size.

374-377: for LDSR, what reference was used for computing LD scores, and what SNPs were included in the LD score regressions? I would also add the genomic control lambda values for each phenotype to Supp. Table 5.

593: Data availability: will the GWAS summary statistics go into an appropriate repository? (allowed under UK Biobank rules)

Reviewer #1 (Remarks to the Author):

In this manuscript, the authors describe the results of a GWAS for 3 depression-related phenotypes from the UK-Biobank data. They report 17 novel hits associated with these phenotypes. Gene-based and pathway analyses highlight the relevance of synaptic mechanisms in these associations. The association data show some correlation with previous studies investigating either clinically defined major depression or a personality trait identified as a risk factor for depression, neuroticism. The strength of the data is the large sample size with uniform phenotyping and genotyping and is currently the largest reported sample for association with depression-related phenotypes (N cases). Overall, the analyses are very solid and convincing. However, there would be possibilities to improve the relevance of the paper for the field.

Response to Reviewer #1 remarks:

Thank you for taking the time to review our submitted manuscript. Your comments have been very valuable and they have improved the quality and impact of the paper.

Reviewer #1, Comment 1:

The authors only compare their data to the data of Hyde et al., in which rather similar questions were used to define depression in somewhat smaller, but still comparable data set (23andme), for single eQTL variants. A more comprehensive comparison to this dataset would be of great interest and could also be used for more formal replication. Currently, no attempt for direct replication has been shown. How well would the data replicate in the 23andme dataset, but also the old PGC dataset with clinical phenotyping? This would also allow to better estimate what the broad depression phenotype means with respect to other phenotyping that has been used before.

Response to Reviewer #1, Comment 1:

This is a very useful suggestion that we have now acted upon in our revised manuscript (line 102 - 106). We contacted 23andMe to obtain their summary statistics and found that the direction of all but one of our 17 significant loci replicated in 23andMe. In addition, we found that ten loci replicated with nominally significant p-values, seven of these met more formal criteria for replication after Bonferroni adjustment. We were unable to test for replication with the 2013 "old PGC" results, as these were imputed to a different panel, and there was limited overlap (5 out of 17) in the variants available. Therefore, this additional analysis wasn't undertaken.

Reviewer #1, Comment 2:

The genetic correlation of 3 phenotypes seems less surprising, given the fact that overlapping phenotypes have been used, so that everybody with the ICD10 diagnosis would also be a case in the 2 other samples. At least this is how I would read the methods. The authors may want to comment on this in the paper.

Response to Reviewer #1, Comment 2:

There is indeed overlap in the individuals that contribute to each depression phenotype, as well as controls that are common to each definition. Nevertheless, the LDSR method adjusts for sample overlap, estimating the degree of genetic architecture after taking into account the overlap between

samples. We have however added further information to the discussion section highlighting this issue, lines 203 - 207. The external comparisons support our conclusion that there is a high genetic correlation between each trait in UKB and an externally and well-validated independent depression trait. Together, these results support the conclusions of our manuscript.

Reviewer #1, Comment 3:

The authors should give a better rationale for why additional gene-based and region-based analyses were run. What is the added information above the GWAS data?

Response to Reviewer #1, Comment 3:

The analysis of only the GWAS data restricts the interpretation of the results to only those variants that pass a particular *P*-value threshold. Interpretations of only GWAS significant loci are also somewhat subjective in nature, and it can be difficult to correctly weight findings that converge on particular cellular or disease processes. The purpose of the additional gene and region-based analyses sought to overcome these difficulties and allow multiple variants, within a gene or region, to be taken in to consideration even where the individual variants may only reach a suggestive level of significance. This effect of genes and pathways can be tested more objectively allowing additional hypotheses to be examined in a relatively unbiased way. We are grateful to the reviewer for raising this issue and we have now updated the text within the paper to make the purpose of conducting such analyses clearer.

Reviewer #1, Comment 4:

The authors explain the lack of overlap of associations across the 3 correlated phenotypes with potential biological differences in each of these phenotypes. This needs to be elaborated, given the overlap of cases. False positive associations could also be a reason for non-overlap. In the two more strict phenotypes, the N of cases drops to 30k and 8k, respectively, sample sizes for which so far no consistent association with depression could be found.

Response to Reviewer #1, Comment 4:

We would like to thank the reviewer for drawing our attention to this issue. We have now included Supplementary Table 2 which provides the effect sizes and *P*-values of the associated variants across all three phenotypes, which is now covered in the discussion, lines 208-218. We have also now extended the paragraph in the discussion that covers this topic to highlight the issue with sample size and power for probable MDD and ICD-coded MDD and also the potential for false positive associations.

Reviewer #1, Comment 5:

It is not clear what figure 3 really adds to the story other than showing that genes in a pathway are correlated. Please explain in the text.

Response to Reviewer #1, Comment 5:

Since reviewer #1 and #2 expressed concerns about the usefulness of Figure 3, it has now been removed from the manuscript.

Reviewer #1, Comment 6:

The discussion reads mostly like a laundry list of genes and their prior associations with other phenotypes. A bigger picture discussion of depression phenotyping, and more detail on what does the association with the broad phenotype could mean for clinical depression would be helpful. This would be especially interesting in the light of replications in other cohort with different levels of phenotyping (self-report vs. clinical).

Response to Reviewer #1, Comment 6:

The section covering the protein coding genes has been moved to the supplementary information with a much more succinct paragraph highlighting potentially important genes identified. We agree with the reviewer's comment that providing a bigger picture of depression phenotyping is an important aspect of the current manuscript that should be made clearer to the reader. In our view, the paper clearly shows that there is value in using very broad low-depth phenotyping to study the genetic architecture of MDD. This has important implications for future studies, where such data could be obtained in large numbers of individuals at lower costs than more clinically robust phenotyping, thus accelerating progress. We have now made this point more clearly in the discussion on lines 241-263.

Reviewer #1, Comment 7:

Overall an interesting paper that could be strengthened by additional analyses and a more thoughtful discussion. Especially an attempt for replication would increase the value of the paper.

Response to Reviewer #1, Comment 7:

Thanks again for your comments which have greatly enhanced the manuscript and we are glad to have been able to include a replication of our findings. The replication analysis, as detailed above, greatly increases confidence in the reported findings and will further extend interest in our published manuscript.

Reviewer #2 (Remarks to the Author):

The authors present GWAS of three depression phenotypes in the latest UK Biobank data. They report a number of associations, mostly with a broad self-reported phenotype of seeking care for depression or related conditions. They also present gene-based, pathway, and eQTL analyses, as well as some analyses of genetic correlations. At a high level the results are consistent with past work, that in this domain broad phenotype definitions with large sample sizes have been more successful for genetic discovery than smaller studies with more rigorous phenotyping.

Response to Reviewer #2 remarks:

Thank you for providing a detailed review of our manuscript. We agree that our findings are consistent with other studies, showing that larger broader phenotyping is more successful than smaller studies with less rigorous phenotyping. There are however several advantages to our work that are novel and important. First, we provide quantitative evidence comparing alternative trait definitions in a single large sample. Secondly, we validate these findings against a well validated independent sample (PGC). We also find new loci and biology for depression and, in responding to both reviewers, provide evidence of replication in 23andMe. We have responded to all of your suggestions below, and as a result we hope you will agree that the manuscript is much improved.

Reviewer #2, Comment 1:

The discussion mentions that several of the reported findings agree with previous studies ([3] and [4]), however, as a whole, the results are presented without replication, and it is hard to assess what is new. Can the authors look up all the index SNPs in any previous study? (maybe inconvenient for [3] since it hasn't been published yet). Also it might be useful to tabulate the GWAS results for each index SNP across the three different phenotypes: it seems worth asking how effect sizes compare for the strict vs broad phenotypes.

Response to Reviewer #2, Comment 1:

We have now tested our reported findings in the next largest single, independent sample (23andMe). We examined the 17 hits that we identified and are pleased to report a high level of replication for the associated variants that we detected using UK Biobank, lines 102-106. To identify novel variants, we have compared our significant variants to the regions (± 500 Kb) containing significant variants identified by the Converge Consortium, the Okbay et al. study and the Hyde et al. analysis of 23andMe. We found that 14 out of our 17 hits are likely to be novel and have included this information in the manuscript, lines 106-110. We have also added an additional supplementary table (Supplementary Table 2) which details the effect sizes and p-values across the three different phenotypes in UK Biobank and extended the discussion to cover this topic.

Reviewer #2, Comment 2:

60-63: I might also mention Okbay et al 2016 as part of the spectrum of depression phenotypes that have been recently studied.

Response to Reviewer #2, Comment 2:

We agree that this would be a useful addition to the paper, and we have now added a more detailed consideration of the Okbay et al. 2016 study.

Reviewer #2, Comment 3:

84: could we also see 2x2 cross tabulation of case/control status for each pair of UK Biobank phenotypes?

Response to Reviewer #2, Comment 3:

We agree that this would be a useful addition to the manuscript. This information has now been added to Supplementary Table 17.

Reviewer #2, Comment 4:

113-124: how can I tell whether these differences are meaningful? Can a test of heterogeneity be done, or a test for regions that differ from the overall average?

Response to Reviewer #2, Comment 4:

We agree that the presence or absence of heterogeneity is not obvious from the figure alone. We have now added a test of heterogeneity of the heritability of the geographical regions to the results, lines 136-138. There was no evidence of heterogeneity for each of the three phenotypes examined.

Reviewer #2, Comment 5:

137: I might note that 11 regions are in the MHC.

Response to Reviewer #2, Comment 5:

To reduce confusion among readers we have removed the identification of regions using the gene-based approach. We have retained the identification of regions based on recombination hot spots, with the MHC region treated as a single region within the 59 significant regions identified for broad depression.

Reviewer #2, Comment 6:

Fig. 3: [editorial digression: I'm not a fan of figures whose interpretation requires me to go read other papers; even after skimming the GeneMania papers I'm not sure I could say what's going on here] What is the diagonal shading? Has the topology been optimized to pull together nodes that were associated with depression? What does "The size of each node is determined by the effect size of the gene coding region within the pathway based on previous studies" mean? (what previous studies?) Why do so many of the nodes have apparently the same size? Are there settings I should know about that affect the results? Why is this figure so nicely spaced out when the examples in the GeneMania paper are all clumpy? Why is there a monotonic progression of circle sizes along the outer ring?

Response to Reviewer #2, Comment 6:

Since reviewer #1 and #2 expressed concerns about the usefulness of Figure 3, it has now been removed from the manuscript.

Reviewer #2, Comment 7:

184-188: the p values for the genetic correlations don't appear in Supp. Table 8 but maybe they should. Also I wonder if the more interesting test is whether these phenotypes are related (i.e. $r_g > 0$) or whether they are different (i.e. $r_g < 1$).

Response to Reviewer #2, Comment 7:

We thank the reviewer for raising this issue. The p-values for all genetic correlations have now been added to the suggested supplementary table. We have also calculated the p-values for $r_g = 1$ and found significant differences between each of the UK Biobank phenotypes. These values have now been added to supplementary table and a more detailed consideration of the main findings has been added to the discussion, lines 198-207.

Reviewer #2, Comment 8:

302: according to [60] there are ~39 million HRC markers, not 19 million? So about 80% have been excluded from analysis, I guess mostly due to the 1% MAF filter? Why was the 1% filter selected? (this excludes many variants with good information scores)

Response to Reviewer #2, Comment 8:

The 19M variants referred to in the manuscript were variants with a MAF > 0.0005 and an imputation accuracy score of > 0.1. We have now updated the reported number of variants in the revised manuscript. We chose to apply a threshold of 1% to our associated analysis to limit false positive associations. Prior analyses of MDD have, in the majority of cases, adopted similar thresholds. The CONVERGE consortium applied a lower MAF threshold using low-pass sequencing, but they too did not report any significant findings below a MAF of 1%. The 23andMe analysis did not report any significant variants beneath this threshold and latest Psychiatric Genomics Consortium analysis of depression (<https://doi.org/10.1101/167577>) also applied a 1% MAF threshold.

Reviewer #2, Comment 9:

307-310: how were call rate and Hardy-Weinberg equilibrium (HWE) filters applied to imputed genotype probabilities or dosages (or am I misinterpreting)? For a sample size of >300K, is a HWE threshold of $1e-6$ rather severe? What proportion of variants are excluded by the HWE filter?

Response to Reviewer #2, Comment 9:

The call rates were provided to us as part of the UK Biobank data release and their documentation indicates these rates were based on genotype probability data. We calculated HWE using PLINK2 which converts Oxford format data files to its native bed/bim/fam format. This native format uses hard called genotypes so we assume HWE is also based on genotype probabilities, although the PLINK2 documentation doesn't specifically address this. There were 87,223 variants excluded based on the applied HWE filter, which was less than 1% of the total number of variants tested. The 10^{-6} threshold is relatively strict, but is in alignment with other analyses of depression and provides greater confidence in the validity of the significant results we have presented. However, we have run our association analysis on these excluded variants and none of them were genome-wide significant ($P < 5 \times 10^{-8}$). We agree that this is an important point, but we hope that this additional information will reassure this reviewer that we haven't lost significant or important findings by adopting this somewhat strict threshold.

Reviewer #2, Comment 10:

353: association tests were performed using linear regression on binary phenotypes after regressing out the covariates. Is linear regression a good solution for this problem? (I suppose it is probably safe for common variants) I might refit logistic regression models, including covariates, for the index SNPs in Table 2, at the least because that would give a more familiar effect size.

Response to Reviewer #2, Comment 10:

We agree that a linear regression is not the conventional solution for modelling binary disease phenotypes. The size (8TB) and format (bgen v1.2) of the UK Biobank dataset required a bespoke statistical package (Bgenie) to enable an association analysis of the dosage data. This package doesn't allow a logistic regression model. We have, in response to this reviewer's comment, extracted the associated variants and run a logistic regression analysis in PLINK to provide the more familiar effect size (odds ratios) which have been added to Supplementary Table 2. The significance of the results from conventional logistic regression and Bgenie are highly correlated ($r=0.987$) with Bgenie tending to give slightly more conservative results.

Reviewer #2, Comment 11:

374-377: for LDSR, what reference was used for computing LD scores, and what SNPs were included in the LD score regressions? I would also add the genomic control lambda values for each phenotype to Supp. Table 5.

Response to Reviewer #2, Comment 11:

The default files described on the LDSR wiki (<https://github.com/bulik/ldsc/wiki/Heritability-and-Genetic-Correlation>) were used to determine the SNPs included in that analysis with the 1000 Genomes European data used as a reference for LD. This information has now been added to the methods section, lines 361-364. The genomic control lambda values have also now been added to supplementary table 9.

Reviewer #2, Comment 12:

593: Data availability: will the GWAS summary statistics go into an appropriate repository? (allowed under UK Biobank rules)

Response to Reviewer #2, Comment 12:

We have made the summary statistics for all three phenotypes available on the Edinburgh DataShare repository and have added this information to the manuscript.

Reviewer #1 (Remarks to the Author):

The authors have addressed all my concerns. The paper has definitely benefited from the added replications and the discussion has much improved in my view.

Reviewer #3 (Remarks to the Author):

This is a study of genetic variants associated with 3 depression-related phenotypes based on the results of a GWAS of $n = 322,580$ persons from the UK-Biobank data. A total of 17 novel genetic loci are identified. Follow up gene-based and pathway analyses suggest synaptic mechanisms as an underlying mechanism. This is a highly interesting study with very important findings. The manuscript has improved during revision process. However, the clarity of the results and the interpretation should be improved. The following issues should be dealt with:

- 1) They have tested for replication in the 23andme dataset, but there is an ongoing study of depression phenotypes from the Psychiatric Genomics Consortium. The results were presented at the recent Psychiatric Genetics Conference. If there are overlapping samples or confirmatory findings, this should be discussed. Further, it would be interesting to have quick look up to test if the recent UKBiobank GWAS results from large scale data mining are in line with the current results: <http://www.nealelab.is/blog/2017/7/19/rapid-gwas-of-thousands-of-phenotypes-for-337000-samples-in-the-uk-biobank> and <https://biobankengine.stanford.edu/search>
- 2) General: the authors should critically revise the text to avoid any confusion about the current broad phenotypes and the clinical diagnosis of major depressive disorder (MDD). More examples of unclear text in the following:

Abstract:

- 3) The wording should be revised to clarify what phenomenon the current depression-related phenotypes are representing. The way the abstract is written may easily be understood as a study of the diagnosis of major depressive disorder. The current GWAS is done on responses from a few questionnaire items and Hospital records, and the relationship to depression diagnosis is not solidly validated. There are many phenotypes that are "found to be significantly genetically correlated with the results from a previous independent study of clinically defined MDD". Thus, this is not specific enough to support claims about MDD, as mentioned in conclusion of abstract: "this disorder". The specification of phenotype does not make the study less interesting and important, but the authors should clarify this issue.

Introduction:

- 4) the phenotypes used in the current study is of major importance, and the question about validation of the UKBiobank questionnaires should be introduced, and the advantages and disadvantages outlined. This is a key question for interpretation of the current results, including the extensive expression and pathway analysis. These results are less relevant if the link to the human phenomenon is not well documented.
- 5) Again, the wording related to current phenotypes and major depressive disorder should be revised to avoid any misunderstandings. They write "depression-related phenotypes within the large UK Biobank cohort, and identify new disease biology based upon our findings" – what DISEASE are they referring to if the questionnaires are not validated as diagnostic tools? Please revise. Further, they have done "(GWAS) of depression" based on 1) "problems with "nerves, anxiety, tension or depression" - which also clearly includes anxiety disorders and 2) "self-reported depressive symptoms" – which is associated with a series of psychiatric disorders 3). The most depression related phenotype is "ICD-10-coded hospital admission records", but little validation is done to support the reliability of hospital records.

Results:

- 6) They discover one locus associated with the most reliable phenotype, ICD-coded MDD. This should be discussed more openly.
- 7) Table 1. they should provide effect size of the replication sample and a meta p-value. The sign

} with normal or bold font is not informative. Please revise and include real numbers or +/- sign.

8) The p-values of the top hits across the 3 UKBiobank phenotypes should be clarified – what does the Supplementary Table 2 show? In general, there seem to be few overlapping top hits based on the Fig 1 a,b,c and Suppl Table 2, but several of the effects go in the same direction across the three different traits? This should be clarified and discussed. Heterogeneity, polygenicity etc

Discussion

9) “This study describes the largest analysis of depression” – I suggest using “depression-related phenotypes”

10) The statement: “However, we also showed that significant differences between the underlying genetic architecture of the three phenotypes and these may be informative for identifying subclasses of depression...” is speculative. The current results be used to define different depression-related phenotypes, not subclasses of depression diagnosis.

11) I appreciate the discussion of phenotypes, but the problem of specificity is not mentioned. How can the authors rule out anxiety disorders in the current dataset? This is important to highlight, as there will be follow up studies (genetic correlations, meta-analysis) building on the current results, and then the lack of specificity should be clearly stated to help interpret the current findings.

12) It seems also evident from the conclusion in the discussion, that the authors are mixing depression-related phenotypes with clinical diagnosis of MDD. The current findings provide...“the discovery of disease mechanisms, pharmacological treatments...” . Based on findings from the broad phenotype, approx 35% of UK population should receive pharmacological treatment for depression (113,769 cases and 208,811 controls) = 23 mill people. This makes no sense, so please revise. Again, such clarifications will only improve the value of the paper and increase the impact.

Methods:

13) The definition of the different depressive phenotypes should be clarified. Supplementary table 17 and the method description of samples should be better explained in relation to the phenotype definitions in supplement.

14) LDSR is not a tool for validating clinical diagnosis of MDD. Further, the authors should provide the specific genetic correlation across different traits, not only MDD: They should investigate if other high powered GWAS of other psychiatric disorders (bipolar disorders, ADHD, ASD and schizophrenia) are significantly different from MDD? The specificity is critical.

15) The Q-Q plots show some inflation, particularly for broad depression phenotype where the lambda gc is 1.32 (Supplementary Table 6). How was this corrected? If not corrected, what is the rationale beyond LDSR intercept? Are the authors suggesting that the broad phenotype is more polygenic than probable MDD? This seems ad hoc based on current presentation and should be supported by objective evidence.

16) The follow-up analysis including the gene-based, pathway, and eQTL analyses are secondary and can be downplayed

Minor

17) 'Abstract – use same number of sample size as in title. No reason to have approx. number in abstract

Reviewer #4 (Remarks to the Author):

Howard et al. use UK Biobank data to perform GWAS for three depression phenotypes, namely broad depression, probable major depressive disorder (MDD) and ICD-coded MDD. They identify and replicate 14 putative novel genome-wide significant loci and perform a number of bioinformatics analyses typically performed on GWAS data. For the broad depression phenotype (the phenotype with most cases, n=113,769), they identify five significantly enriched gene sets pointing towards relatively generic cellular components of the nervous system.

While the novel variants for depression are highly relevant for further molecular and epidemiological studies of depression, the biological pathway results could be described in higher detail to further enhance the significance of this paper. In its current form, the novelty of the paper seems to be on the newly discovered loci rather than elucidation of biological pathways etiologic to depression. The following points could probably improve the paper:

- The authors' explicitly note that their work shows that excitatory synapses play a role in the pathology of depression. However, the role of excitatory synapses in depression seems, already to be an established hypothesis that is actively discussed in this field. The authors should be more specific on how their results further the current understanding on the role of excitatory synapses in depression.
- The authors should clearly state the Gene Ontology gene set identifiers of these enriched gene sets and report the gene set genes that overlapped with GWAS loci. Without this information it is hard to fully understand the molecular implications of their gene set findings. Also, in the supplementary data they authors' should report on overlaps between the gene sets. Further they should mention how to interpret the beta variable in the gene set results table (Tab. 2). Finally, should the authors decide not to include any gene set figure, then they should remove the following sentence from the Methods: "Visualisation of pathways was obtained using the online tool, GeneMANIA32").
- It would substantially add to their work if the authors could integrate publicly available brain single cell transcriptomics data sets with their GWAS data (e.g. Roly Poly, MAGMA or DEPICT). Any indication brain areas and cell types that express genes from associated depression loci would be relevant. Should this analysis turn out negative, the authors could, based on the existing single cell transcriptomics data, report in which brain areas and cell types the prioritized genes and gene sets genes seem to be expressed.
- The authors should state whether the genes prioritized for MAGMA are driven by multiple independent variants or whether this analysis primarily aims at identifying a larger number of tentatively associated genes (compared to those in associated loci) due to the reduced the multiple correction burden. Also, it should be noted that several genes probably are scored by the same SNP(s) and hence should not be regarded as independent findings (e.g. the histone and zink finger cluster genes on chromosome 6).
- Did the eQTL data provide target genes that overlap with or provide further evidence for the five enriched gene sets?

Reviewer #1 (Remarks to the Author):

The authors have addressed all my concerns. The paper has definitely benefited from the added replications and the discussion has much improved in my view.

Response to reviewer #1:

Thank you for reviewing our manuscript, and for confirming that our revisions have addressed all of your concerns. We feel the paper has improved greatly as a result of the comments provided by all reviewers.

Reviewer #3 (Remarks to the Author):

This is a study of genetic variants associated with 3 depression-related phenotypes based on the results of a GWAS of $n = 322,580$ persons from the UK-Biobank data. A total of 17 novel genetic loci are identified. Follow up gene-based and pathway analyses suggest synaptic mechanisms as an underlying mechanism. This is a highly interesting study with very important findings. The manuscript has improved during revision process. However, the clarity of the results and the interpretation should be improved.

Response to reviewer #3:

Thank you for taking the time to read and provide comments on our manuscript. We have now addressed your comments and suggestions below, and we hope that the paper is now much stronger based on these changes and has greater phenotypic clarity.

Reviewer #3, comment #1:

They have tested for replication in the 23andme dataset, but there is an ongoing study of depression phenotypes from the Psychiatric Genomics Consortium. The results were presented at the recent Psychiatric Genetics Conference. If there are overlapping samples or confirmatory findings, this should be discussed. Further, it would be interesting to have quick look up to test if the recent UKBiobank GWAS results from large scale data mining are in line with the current results:

<http://www.nealelab.is/blog/2017/7/19/rapid-gwas-of-thousands-of-phenotypes-for-337000-samples-in-the-uk-biobank> and <https://biobankengine.stanford.edu/search>

Response to reviewer #3, comment #1:

We are aware of the recent analysis conducted by the Psychiatric Genomics Consortium and we are very excited about their results. Their paper is not yet published and so the results are not currently available. We also understand that they have included both the 23andMe discovery cohort and an earlier release of the UK Biobank data. Therefore, any attempt at replication in the PGC would be confounded by sample overlap.

A large amount of time and effort was spent by clinical psychiatrists located at University of Edinburgh, King's College London and University of Glasgow in determining the phenotypes that were used in this study. We do not think it would be appropriate to re-examine our results using the analysis conducted by the Neale lab based on single fields, as these do not incorporate some of the disease exclusions needed for the current analysis, such as the presence of schizophrenia or bipolar disorder.

Reviewer #3, Comment #2:

General: the authors should critically revise the text to avoid any confusion about the current broad phenotypes and the clinical diagnosis of major depressive disorder (MDD). More examples of unclear text in the following:

Response to reviewer #3, comment #2:

Having reread the manuscript, the authors agree that there was ambiguity in the wording used throughout the manuscript. We have now taken great care to ensure that the wording reflects the phenotypes we are examining. The text now clearly highlights the differences between our phenotypes (lines 69-71, 229-231, 275-286 and 340-388) and we hope this will help other researchers consider which phenotype is appropriate for any additional analyses that follow our work.

Reviewer #3, Comment #3:

Abstract:

3) The wording should be revised to clarify what phenomenon the current depression-related phenotypes are representing. The way the abstract is written may easily be understood as a study of the diagnosis of major depressive disorder. The current GWAS is done on responses from a few questionnaire items and Hospital records, and the relationship to depression diagnosis is not solidly validated. There are many phenotypes that are “found to be significantly genetically correlated with the results from a previous independent study of clinically defined MDD”. Thus, this is not specific enough to support claims about MDD, as mentioned in conclusion of abstract: “this disorder”. The specification of phenotype does not make the study less interesting and important, but the authors should clarify this issue.

Response to reviewer #3, comment #3:

We have revised the wording in the abstract (lines 37 and 45) and removed the sentence regarding the genetic correlation with the previous study of MDD. We have also removed “this disorder” from the last sentence of the abstract. We have also screened the manuscript for further ambiguities (including lines 63, 206, 230, 287 and 307) and we hope that the reviewer will find the text much improved as a result.

Reviewer #3, Comment #4:

Introduction:

4) the phenotypes used in the current study is of major importance, and the question about validation of the UKBiobank questionnaires should be introduced, and the advantages and disadvantages outlined. This is a key question for interpretation of the current results, including the extensive expression and pathway analysis. These results are less relevant if the link to the human phenomenon is not well documented.

Response to reviewer #3, comment #4:

The introduction has been amended to highlight key differences (lines 65-71) between the phenotypes. The section in the discussion covering the interpretation of the phenotypes has also been revised (lines 274-305) to provide a more in-depth examination of the phenotypes used and provides the advantages and disadvantages of each. The method section of our manuscript (lines 339-388) has also been extended further in this respect, to improve clarity and facilitate replication and further analysis.

Reviewer #3, Comment #5:

5) Again, the wording related to current phenotypes and major depressive disorder should be revised to avoid any misunderstandings. They write “depression-related phenotypes within the large UK Biobank cohort, and identify new disease biology based upon our findings” – what DISEASE are they referring to if the questionnaires are not validated as diagnostic tools? Please revise. Further, they have done “(GWAS) of depression” based on 1) “problems with “nerves, anxiety, tension or depression” - which also clearly includes anxiety disorders and 2) “self-reported depressive symptoms” – which is associated with a series of psychiatric disorders 3). The most depression related phenotype is “ICD-10-coded hospital admission records”, but little validation is done to support the reliability of hospital records.

Response to reviewer #3, comment #5:

Thanks once again for highlighting areas of the text that are misleading. In response to this comment, we have now removed the sentence that talks about “disease biology”. We have also added additional information to the phenotype section within the methods (lines 339-388). Taking in to account sub points 1 and 2, we have also gone through the entire manuscript and highlighted where appropriate the extent of the broad depression phenotype and its likely overlap with related disorders (including lines 69-71, 275-282, 311-312 and 352-354). To support the reliability of the hospital records we also examined whether those classes as ICD-coded cases also had reported additional instances of depression. We found that over 90% of the ICD-coded MDD cases had either seen a doctor or psychiatrist for nerves, anxiety, tension or depression or had been depressed/down or suffered from anhedonia for a whole week. This additional validation has been added to the methods (lines 384-388).

Reviewer #3, Comment #6:

Results:

6) They discover one locus associated with the most reliable phenotype, ICD-coded MDD. This should be discussed more openly.

Response to reviewer #3, comment #6:

We have now added to the discussion highlighting that just one significant variant was found for ICD-coded MDD (lines 393-397).

Reviewer #3, Comment #7:

7) Table 1. they should provide effect size of the replication sample and a meta p-value. The sign) with normal or bold font is not informative. Please revise and include real numbers or +/- sign.

Response to reviewer #3, comment #7:

We have revised table 1 to include the effect size observed within 23andMe and removed the bold highlighting. We have also run a meta-analysis of the two cohorts and included the effect sizes, p-values and +/- signs.

Reviewer #3, Comment #8:

8) The p-values of the top hits across the 3 UK Biobank phenotypes should be clarified – what does the Supplementary Table 2 show? In general, there seem to be few overlapping top hits based on the Fig 1 a,b,c and Suppl Table 2, but several of the effects go in the same direction across the three different traits? This should be clarified and discussed. Heterogeneity, polygenicity etc

Response to reviewer #3, comment #8:

We agree that there seems to be few overlapping top hits and a discussion of supplementary table 2 is made on lines 231-241. There is extensive, but not complete, sample overlap between each of the three phenotypes and therefore assessments of heterogeneity and polygenicity are likely to be confounded. Therefore, the authors think it more appropriate to simply provide readers with the findings and highlight any differences. Non-overlap in the top genome-wide significant hits is not, in our opinion, surprising until GWAS studies begin to approach sample sizes with the power to identify all associated variants.

Reviewer #3, Comment #9:

Discussion

9) "This study describes the largest analysis of depression" – I suggest using "depression-related phenotypes"

Response to reviewer #3, comment #9:

We agree, and this has now been updated and we have incorporated this throughout the manuscript (including lines 37, 63, 158, 206, 230, 307, 437).

Reviewer #3, Comment #10:

10) The statement: "However, we also showed that significant differences between the underlying genetic architecture of the three phenotypes and these may be informative for identifying subclasses of depression...." is speculative. The current results be used to define different depression-related phenotypes, not subclasses of depression diagnosis.

Response to reviewer #3, comment #10:

Thank you for highlighting this and it has now been corrected in the manuscript (lines 229-231).

Reviewer #3, Comment #11:

11) I appreciate the discussion of phenotypes, but the problem of specificity is not mentioned. How can the authors rule out anxiety disorders in the current dataset? This is important to highlight, as there will be follow up studies (genetic correlations, meta-analysis) building on the current results, and then the lack of specificity should be clearly stated to help interpret the current findings.

Response to reviewer #3, comment #11:

We have updated the discussion (lines 277-278) to highlight that the lack of specificity of the broad depression phenotype. Additionally, we have calculated and included in the discussion (lines 278-282) the genetic correlation of anxiety disorder (using the factor scores for anxiety by Otowa et al.: <https://www.nature.com/articles/mp2015197>) with our depression-related phenotypes. The broad depression phenotype did not have a markedly greater genetic correlation ($r_g = 0.52 \pm 0.11$, $p = 4.5e-6$) compared to either probable MDD ($r_g = 0.60 \pm 0.17$, $p = 0.0005$) or ICD-coded MDD ($r_g = 0.47 \pm 0.18$, $p = 0.008$). Nevertheless, our cross-trait depression genetic correlation findings empirically demonstrate that the overlap in the common genetic architecture of broad and narrowly defined depression is substantial.

Reviewer #3, Comment #12:

12) It seems also evident from the conclusion in the discussion, that the authors are mixing depression-related phenotypes with clinical diagnosis of MDD. The current findings provide....“the discovery of disease mechanisms, pharmacological treatments...” . Based on findings from the broad phenotype, approx 35% of UK population should receive pharmacological treatment for depression (113,769 cases and 208,811 controls) = 23 mill people. This makes no sense, so please revise. Again, such clarifications will only improve the value of the paper and increase the impact.

Response to reviewer #3, comment #12:

We have now reviewed the entire manuscript and ensured that we are not mixing our derived depression phenotypes with a Structured Clinical Interview for DSM diagnosis (including lines 37, 63, 158, 206, 230, 307, 437).

Reviewer #3, Comment #13:

Methods:

13) The definition of the different depressive phenotypes should be clarified. Supplementary table 17 and the method description of samples should be better explained in relation to the phenotype definitions in supplement.

Response to reviewer #3, comment #13:

We have expanded the different phenotypic descriptions in the methods (lines 339-390) and we feel that it is now much clearer. We have also incorporated additional information from supplementary table 17 (lines 367-370 and 380-384) which aids in the interpretation of the differences between the phenotypes. We also have now highlighted potential overlap with other disorders (including lines 69-71, 275-282, 311-312 and 352-354) and included an attempt at validating the hospital records using the two other phenotypes (lines 384-388).

Reviewer #3, Comment #14:

14) LDSR is not a tool for validating clinical diagnosis of MDD. Further, the authors should provide the specific genetic correlation across different traits, not only MDD: They should investigate if other high powered GWAS of other psychiatric disorders (bipolar disorders, ADHD, ASD and schizophrenia) are significantly different from MDD? The specificity is critical.

Response to reviewer #3, comment #14:

We have now removed the text that suggests that genetic correlations are able to validate clinical diagnosis. We had not intended to provide this analysis as a test of validity, simply one that demonstrated substantially overlapping common genetic architecture. We have also now included additional analysis using LD Hub and reported genetic correlations across 235 other traits with our depression-related traits (lines 154-166, 221-227 and 441-443). We have focussed the paper on covering the correlations with the psychiatric traits as suggested by the reviewer.

Reviewer #3, Comment #15:

15) The Q-Q plots show some inflation, particularly for broad depression phenotype where the lambda gc is 1.32 (Supplementary Table 6). How was this corrected? If not corrected, what is the rationale beyond LDSR intercept? Are the authors suggesting that the broad phenotype is more polygenic than probable MDD? This seems ad hoc based on current presentation and should be supported by objective evidence.

Response to reviewer #3, comment #15:

We found no evidence of residual population stratification using the intercept and standard errors obtained from LDSR and so no correction was required. We have updated the method section (lines 413-416) to make it clearer that the observed inflation was due to polygenic signal and not population structure. We have similarly updated the main text (lines 119-123) to highlight that inflation in lambda gc is driven by polygenic signal. It is likely that the broad phenotype has a greater polygenic signal, most likely due to increased power. However, in response to your comment, we have decided to detail the observed values from the LDSC analysis to allow the reader to form their own opinion.

Reviewer #3, Comment #16:

16) The follow-up analysis including the gene-based, pathway, and eQTL analyses are secondary and can be downplayed

Response to reviewer #3, comment #16:

The editor and reviewer #4 have indicated that they would more attention placed on the biological implications of the findings. We understand the reviewer's point of view, but we have had to respond to the concerns of other reviewers on this issue.

Reviewer #3, Comment #17:

Minor

17) 'Abstract – use same number of sample size as in title. No reason to have approx. number in abstract

Response to reviewer #3, comment #17:

Agreed, we have updated the sample size in the abstract (line 36) accordingly.

Reviewer #4 (Remarks to the Author):

Howard et al. use UK Biobank data to perform GWAS for three depression phenotypes, namely broad depression, probable major depressive disorder (MDD) and ICD-coded MDD. They identify and replicate 14 putative novel genome-wide significant loci and perform a number of bioinformatics analyses typically performed on GWAS data. For the broad depression phenotype (the phenotype with most cases, n=113,769), they identify five significantly enriched gene sets pointing towards relatively generic cellular components of the nervous system.

While the novel variants for depression are highly relevant for further molecular and epidemiological studies of depression, the biological pathway results could be described in higher detail to further enhance the significance of this paper. In its current form, the novelty of the paper seems to be on the newly discovered loci rather than elucidation of biological pathways etiologic to depression. The following points could probably improve the paper:

Response to reviewer #4:

The co-authors would like to thank reviewer #4 for reading the manuscript and providing insightful comments which have greatly improved the manuscript.

Reviewer #4, Comment #1:

The authors' explicitly note that their work shows that excitatory synapses play a role in the pathology of depression. However, the role of excitatory synapses in depression seems, already to be an established hypothesis that is actively discussed in this field. The authors should be more specific on how their results further the current understanding on the role of excitatory synapses in depression.

Response to reviewer #4, comment #1:

We have now updated the discussion (lines 256-259) to include additional papers that discuss the role of excitatory synapses in depression. We have also extended the discussion (lines 259-262 and 271-273) to highlight the potential of a genome-wide association study approach to identify the important genes in each pathway for the respective phenotypes. However, we would argue that demonstrating these findings using GWAS is far from redundant. The convergence of findings provides substantial further confidence in previous findings. GWAS also has several advantages over other methods, not least of which are its lesser susceptibility to confounding and reverse causation.

Reviewer #4, Comment #2:

The authors should clearly state the Gene Ontology gene set identifiers of these enriched gene sets and report the gene set genes that overlapped with GWAS loci. Without this information it is hard to fully understand the molecular implications of their gene set findings. Also, in the supplementary data they authors should report on overlaps between the gene sets. Further they should mention how to interpret the beta variable in the gene set results table (Tab. 2). Finally, should the authors decide not to include any gene set figure, then they should remove the following sentence from the Methods: "Visualisation of pathways was obtained using the online tool, GeneMANIA32").

Response to reviewer #4, comment #2:

We have now added the Gene Ontology gene set identified to Table 2 and also added additional supplementary tables containing the full gene sets within each of the enriched pathways (Supplementary Table 17) as well as a contingency table (Supplementary Table 18) that illustrates the overlap between the different pathways. We have highlighted the extent of overlap in the discussion (lines 255-256). We have added details to the methods (lines 470-472) on the calculation of the beta values for the pathways, and from our understanding the beta values are not directly interpretable as an effect of the pathways. We have removed the sentence regarding the visualisation of the pathways.

Reviewer #4, Comment #3:

It would substantially add to their work if the authors could integrate publicly available brain single cell transcriptomics data sets with their GWAS data (e.g. Roly Poly, MAGMA or DEPICT). Any indication brain areas and cell types that express genes from associated depression loci would be relevant. Should this analysis turn out negative, the authors could, based on the existing single cell transcriptomics data, report in which brain areas and cell types the prioritized genes and gene sets genes seem to be expressed.

Response to reviewer #4, comment #3:

This is an important and helpful suggestion and an analysis we were keen to undertake in response to this comment. We have now added an analysis of tissue types looking for enrichment of variants

involved in those pathways using DEPICT. The results of that analysis were null but it was interesting to note that the most enriched tissues for broad depression were related to the brain and nervous system. This analysis has now been added to the manuscript (lines 191-196 and 474-478).

Reviewer #4, Comment #4:

The authors should state whether the genes prioritized for MAGMA are driven by multiple independent variants or whether this analysis primarily aims at identifying a larger number of tentatively associated genes (compared to those in associated loci) due to the reduced the multiple correction burden. Also, it should be noted that several genes probably are scored by the same SNP(s) and hence should not be regarded as independent findings (e.g. the histone and zink finger cluster genes on chromosome 6).

Response to reviewer #4, comment #4:

We have now updated the method section (lines 446-448) relating to MAGMA to highlight that the MAGMA package accounts for the LD between variants and multi-marker effects. You are right in that several genes are likely to be scored by the same SNP, but this should only occur where the SNP is in LD and influences multiple genes. Our paper seeks to highlight potential genes associated with our depression-related phenotypes and other groups can potentially employ animal models to directly assess the effect of each gene.

Reviewer #4, Comment #5:

Did the eQTL data provide target genes that overlap with or provide further evidence for the five enriched gene sets?

Response to reviewer #4, comment #5:

We examined the genes highlighted by the expression analysis within the enriched gene-sets and there was no overlap and we have updated the results accordingly (lines 195-196).

Reviewer #3 (Remarks to the Author):

The paper has improved significantly

Reviewer #4 (Remarks to the Author):

The authors have addressed all my concerns. Thank you for this valuable contribution.